# Cortical-striatal brain network distinguishes deepfake from real speaker identity
Claudia Roswandowitz [1,2,3] ✉, Thayabaran Kathiresan[4,5], Elisa Pellegrino[2], Volker Dellwo[2] & Sascha Frühholz[1,3,6]

Deepfakes are viral ingredients of digital environments, and they can trick human cognition into misperceiving the fake as real. Here, we test the neurocognitive sensitivity of 25 participants to accept or reject person identities as recreated in audio deepfakes. We generate high-quality voice identity clones from natural speakers by using advanced deepfake technologies. During an identity matching task, participants show intermediate performance with deepfake voices, indicating levels of deception and resistance to deepfake identity spoofing. On the brain level, univariate and multivariate analyses consistently reveal a central cortico-striatal network that decoded the vocal acoustic pattern and deepfake-level (auditory cortex), as well as natural speaker identities (nucleus accumbens), which are valued for their social relevance. This network is embedded in a broader neural identity and object recognition network. Humans can thus be partly tricked by deepfakes, but the neurocognitive mechanisms identified during deepfake processing open windows for strengthening human resilience to fake information.

With the rapid evolution of artificial intelligence technologies, there has been a massive increase in the creation and dissemination of so-called deepfakes. Deepfakes concern synthetic re-creations of social events and/or human features to the degree that recipients of deepfake information would appraise it as real and trustworthy[1]. Deepfakes target sensitive personal information, as well as events and persons of public relevance[2], which poses the risk of abuse in political (e.g. spreading false information) and criminal contexts (e.g., online-banking verification by fake voice identities). Nonetheless, there are also positive applications of deepfake technologies, such as in medical treatments (e.g. synthetic voice profiles for laryngeal patients) and digital communication (e.g. individual voice dubbing).

Human person identity is one of the major targets of deepfake cloning[3]. Identity is expressed in person-specific visual and acoustic features. Concerning the latter, each human has a unique voice profile, which other humans[4] and automatic systems[5] use to identify the individual person. Most recent voice synthesizing algorithms have become powerful, allowing the creation of deepfake clones that mimic identity features of natural speakers with a high level of quality and similarity[6]. Although a large research effort is underway to develop computer algorithms for automatic deepfake generation and detection, little is known about the human ability to reliably recognize socially relevant identity information in audio[7–11] and visual[12–14] deepfakes. This is an important social and societal question, as such deepfakes can infiltrate and disrupt social cohesion and trust within and across social groups[15]. Perceptually, human detection of audio deepfake identity as fake is unstable, across different voice synthesizing algorithms[7,8,11], and even when participants were introduced to the deepfake manipulation[8]. Knowledge about the neuronal mechanisms involved in the processing of cloned speaker identities is missing to date. We are here using psychoacoustical methods to test how well human voice identity is preserved in deepfake voices. As well as neuroimaging methods to investigate the human neurocognitive system recruited when accepting (as a potential neural indicator of deepfake deception) or rejecting (as a potential neural indicator of deepfake resilience) the deepfake as a match for a natural human identity. For this purpose, we generated deepfake voice identities, familiarized participants with the natural identities, and quantified behavioral and neural measures during an identity task in which participants classified person identities from natural and deepfake voices. Since current deepfake creation algorithms are not yet able to perfectly replicate the corresponding natural acoustic identities given some remaining synthesis artifacts, we included a control task including the identical acoustic stimuli but asked participants to

[1]Cognitive and Affective Neuroscience Unit, Department of Psychology, University of Zurich, Zurich, Switzerland. [2]Phonetics and Speech Sciences Group, Department of Computational Linguistics, University of Zurich, Zurich, Switzerland. [3]Neuroscience Centre Zurich, University of Zurich and ETH Zurich, Zurich, Switzerland. [4]Centre for Neuroscience of Speech, University Melbourne, Melbourne, Australia. [5]Redenlab, Melbourne, Australia. [6]Department of Psychology, University of Oslo, Oslo, Norway. ✉e-mail: claudia.roswandowitz@uzh.ch

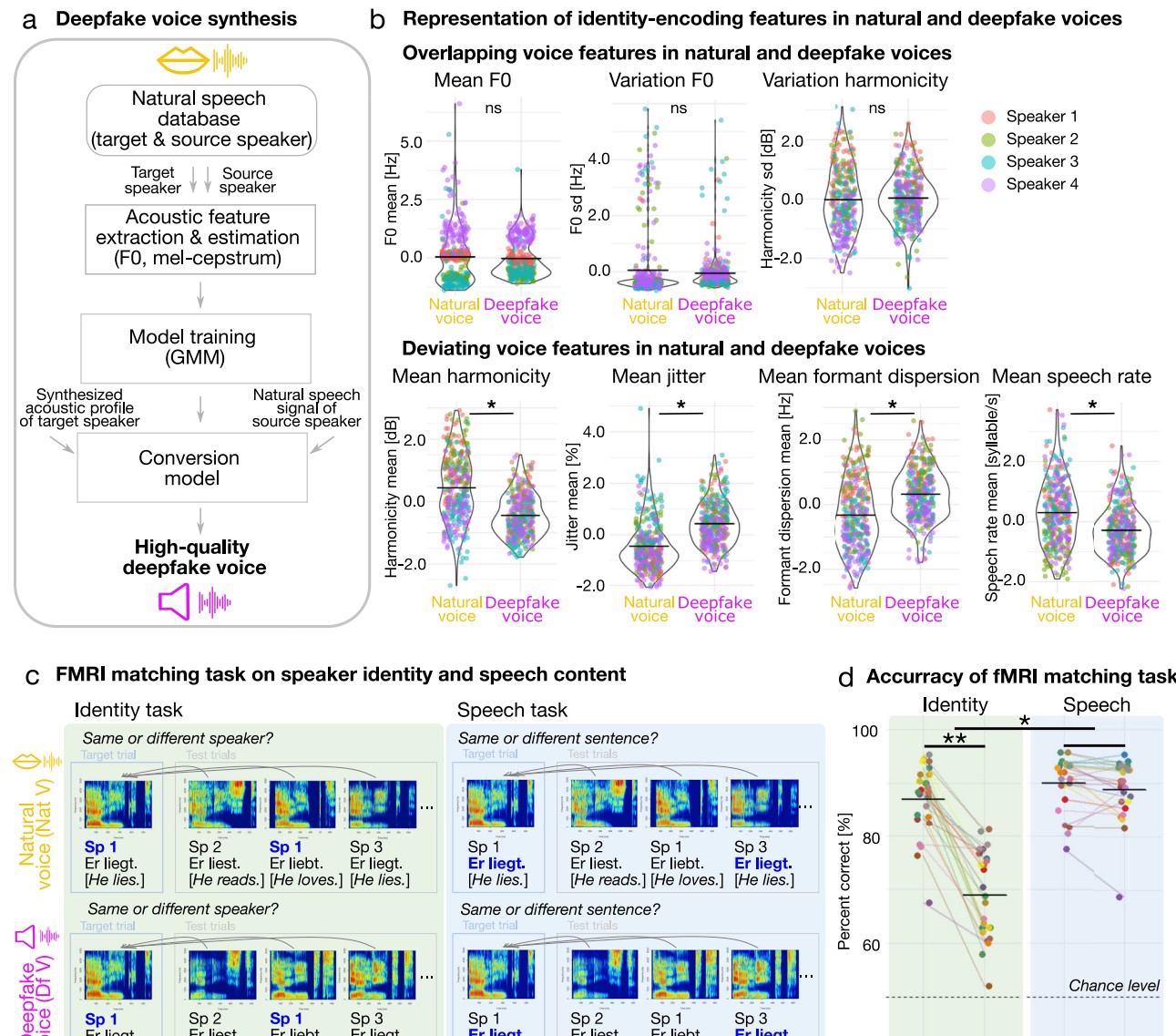

**Fig. 1 | Overview of the deepfake voice synthesis, acoustic representation of identity-encoding features, and experimental tasks. a** The deepfake synthesis consisted of the acoustic voice feature extraction of the natural target and source speaker, training of the Gaussian mixture model (GMM), and conversion of the synthesized idiosyncratic acoustic voice profile of the target speaker with the natural speech sound of the source speaker. **b** Distribution of identity-encoding voice features in natural and deepfake voices. We scaled acoustic values to facilitate visualization and model output comparisons. LMMs assessed acoustic differences between natural and deepfake voices. Asterisks indicate $*p < .05$, Bonferroni-corrected for $n = 7$ models, $ns$: nonsignificant. Circles indicate sentence-specific acoustic values with speaker-specific color coding. Horizontal lines indicate the mean values. **c** Experimental design of the fMRI matching task, including an identity and speech task. **d** Accuracy of the fMRI matching task. Statistics based on LMM ($n_{observations}$=100, $n_{participants}$ = 25) with task and sound condition as fixed effects and participants as a random factor. Asterisks indicate $**p < .0001$, $*p < .001$. Circles indicate individual data and horizontal lines mean performances.

perform an orthogonal speech task. With this control task, we were able to statistically control for potential acoustical confounds originating from the voice deepfake synthesis. This enabled us to separate bottom-up acoustic effects from more abstract top-down effects, both of which could play a role in detecting deepfake identities.

## Results

### Common and differential acoustic features in deepfake voices for human identity

For the deepfake voice synthesis, we recorded the voices of four natural and unfamiliar male speakers (83 German two-word sentences, natural voice) and used an open-source voice conversion algorithm[16], resulting in deepfake utterances (same set of 83 German two-word sentences, deepfake voices) that cloned the idiosyncratic acoustic profile of the four

speakers (Fig. 1a). The voice conversion procedure was based on a Gaussian mixture model and resulted in high-quality deepfake voice samples[17]. Natural and deepfake voices are available here https://doi.org/10.17605/OSF.IO/89M2S[18].

As a first analysis, we assessed how well the acoustic representation of natural voice identities is preserved in the corresponding deepfake instances. We performed an acoustic analysis on seven acoustic features that typically encode voice identity[19]. Our data showed overlapping acoustic profiles (i.e. nonsignificant difference; linear mixed models (LMMs), $n = 7$, Bonferroni-corrected) across the natural and deepfake voices for the mean (estimate = $-0.07$, 95% confidence interval (CI) $[-0.17;0.03]$, $p = 0.174$) and variation of the fundamental frequency (F0, estimate = $-0.11$, CI$[-0.26;0.04]$, $p = 0.145$), which mainly contributes to the perception of vocal pitch. No difference was found for the variation of vocal harmonicity

representing voice-quality dynamics (estimate=0.05, CI[−0.06;0.17], $p = 0.371$) (Fig. 1b, Supplementary Table 1).

Central acoustic features of voice identity thus seem to be preserved in deepfake voices. However, natural and deepfake voices significantly differed in other acoustic features, such as formant dispersion representing vocal timbre (estimate=0.65, CI[0.57;0.72], $p < 0.001$), voice jitter representing natural micro-fluctuations of vocal pitch (estimate=0.88, CI[0.76;0.99], $p < 0.001$), mean of vocal harmonicity as a proxy of vocal sonority (estimate = −0.89, CI[−0.97;−0.80], $p < 0.001$), and speech rate representing vocalization flow and rhythmicity (estimate = −0.60, CI[−0.69;-0.51], $p < 0.001$) (Fig. 1b, Supplementary Table 1). Thus, unlike the commonalities, there seem to be considerable acoustic differences between natural and deepfakes voices, which should enable humans to detect deepfake identities to some degree.

## Humans partly accept deepfake voice identities as real

Before running the main experiment on identity perception of such deepfake voices, we ensured that participants successfully established robust generative models of our four natural speaker identities. This was done in a speaker familiarization task in which participants learned the speakers´ natural voices in association with the synchronized dynamic faces and the written names (Supplementary Fig. 1a). As audio deepfakes aim to replicate the identity information of the original natural speaker, the associated biographical information should be available for natural and deepfake voice identities. All participants with above 80% learning accuracy (mean=96.83%, range=87.50–100%) and thus successful speaker familiarization (Supplementary Fig. 1b) were invited to the neuroimaging session in the next step of the study. We measured brain activity with functional magnetic resonance imaging (fMRI) while participants performed an identity and speech-matching task in alternating blocks, in which we presented either natural or deepfake voices (Fig. 1c). In each task block, participants were presented with a natural target utterance, followed by a sequence of 12 test utterances. In the identity-matching task, participants had to decide whether the test voice matched the target identity (same/different decision).

For the natural voice condition, in which we presented test utterances from natural voices, participants had a high accuracy in identity matching (mean=86.98%, SD = 6.64, Fig. 1d). In the deepfake voice condition, we again presented a natural target voice, but all test voices were deepfake identities. Identity matching was significantly decreased ($N = 25$, emmeans post hoc analysis, estimate=18.05, CI[15.35;20.75], $p < 0.0001$, Supplementary Table 2) for the deepfake (mean=68.94%, SD = 8.15) compared to the natural voice condition (Fig. 1d). This indicates a non-perfect match in identity between the natural and deepfake voices, likely related to the deviating voice features, prompting mismatch decisions with equally frequent miss (28.64%) and false-alarm responses (29.86%) (Supplementary Table 3). Notably, in almost 70% of the trials, deepfake-to-natural identity matching was correct and thus well above the 50% chance level, which we think is enabled by the well-preserved speaker-specific distribution of the natural F0 (Fig. 1d).

To test whether the difference in the natural and deepfake identity task was specific to identity information or related to basic acoustic differences between natural and deepfake voices, we analyzed the behavioral data from the independent speech task (Fig. 1c). Participants were listening to the same stimulus material but were asked to match the linguistic content between a test and target utterance (i.e. same/different linguistic content), again during both the natural and the deepfake voice condition. Unlike in the identity task, accuracies in the speech task were overall high and comparable for the natural (90.07%, SD = 5.03%) and deepfake voices (88.74%, SD = 5.66%; emmeans post-hoc analysis: estimate=1.33, CI[-1.37;4.03], $p = .33$, Fig. 1d, Supplementary Table 2). The double dissociation in identity and speech-matching decisions was confirmed by a significant task-by-naturalness interaction, suggesting that performance differences observed for the identity task (LMM: estimate=16.72, CI[12.96;20.47], $p < .001$, Fig. 1d, Supplementary Table 2) is task specific and not related to basic perceptual

differences between natural and deepfake voices. The next step was to test whether performance differed in the identity and speech tasks presenting natural voices. The natural task conditions served as baseline conditions to determine whether the difficulty of the identity and speech task was comparable. Statistically, performance in the natural speech task was superior to performance in the natural identity task (Percent correct [only natural voices] ~ Task + (1 | VP): estimate: 3.09, CI[1.34;4.84], $p$ value: .001). However, relative performance was high in both tasks (natural identity: 86.98% (+-6.64), natural speech: 90.07 ( + -5.02)) and the difference in percentage correct was only 3.09%. Moreover, six participants performed better in the natural identity compared to the natural speech task (Supplemental Fig. 2). Given the marginal difference in percent correct and the participants who performed better in the identity task, we argue from a perceptual perspective that the difficulties of the identity and speech task were relatively well matched.

Although participants were not informed about their performance during the task, we were interested in whether they nevertheless processed incorrect decisions. We quantified the phenomenon of post error slowing as a measure of cognitive error processing[20,21]. Reaction time differences between post- and pre-error trials did not differ between task and voice condition (LMM, task-by-naturalness interaction, estimate = -353.52, CI[-1131.99;424.96], $p < .373$, Supplementary Table 4) and did not slow down during the deepfake identity task (One Sample t-test, t = 0.269, $p = .394$, Supplementary Table 4), that was associated with the highest number of erroneous trials.

## Ventral striatum separates natural from deepfake voices for identity recognition

Since the identity-matching performance indicated that humans are capable of perceptually detecting deepfake instances of voice identity to a certain degree, we next tested how the human brain, which is assumed to be evolutionarily optimized to register and discriminate natural voice identities, processes and potentially detects deepfake identities.

For the functional brain data, we found higher activity for the natural than for the deepfake voices during the identity task in the bilateral nucleus accumbens (NAcc) as part of the ventral striatum, left anterior cingulate gyrus, bilateral frontal pole, and left posterior middle temporal gyrus (Supplementary Fig. 3a, b, Supplementary Table 5). Of these regions, we observed a significant task-by-naturalness interaction in the right NAcc (Fig. 2b, Supplementary Table 5). The interaction was driven by reduced NAcc activity during the deepfake identity task compared with all other task conditions. This specifically also meant that the data showed increased NAcc activity for the identity task presenting natural voices, as well as the speech task presenting natural and deepfake voices. All task conditions with high matching performance. Thus, univariate findings suggest that the NAcc is sensitive to items that are rewarding for the ongoing task[22–24]. During the natural identity task, voices closely resemble the identity of the previously familiarized speakers and thus seem to more successfully guide identity-matching decisions than the deepfake version of the speakers' identity. For the speech task, natural and deepfake voices seem to convey the task-relevant information equally well (Fig. 1d). For completeness of data analysis, no significant difference in brain activity was found for the speech task when contrasting natural and deepfake conditions.

## Auditory cortex is sensitive to deepfake voice information independent of the task

When testing for brain regions being more responsive to deepfake in contrast to natural voice identities, we found activity across the bilateral voice-sensitive primary AC (pAC) and higher-order AC (hAC) (Fig. 2a), with maximum peaks in the anterior superior temporal gyrus (STG) (Supplementary Table 5), which is central for neural voice-identity processing[4,25,26]. Unlike the NAcc activity, activity in the pAC and hAC did not survive a task-by-naturalness interaction analysis (Fig. 2b), implying more task-general effects in the AC (Supplementary Fig. 3a and c, Supplementary Tables 5 and 6), potentially encoding the artificial and mismatched nature of

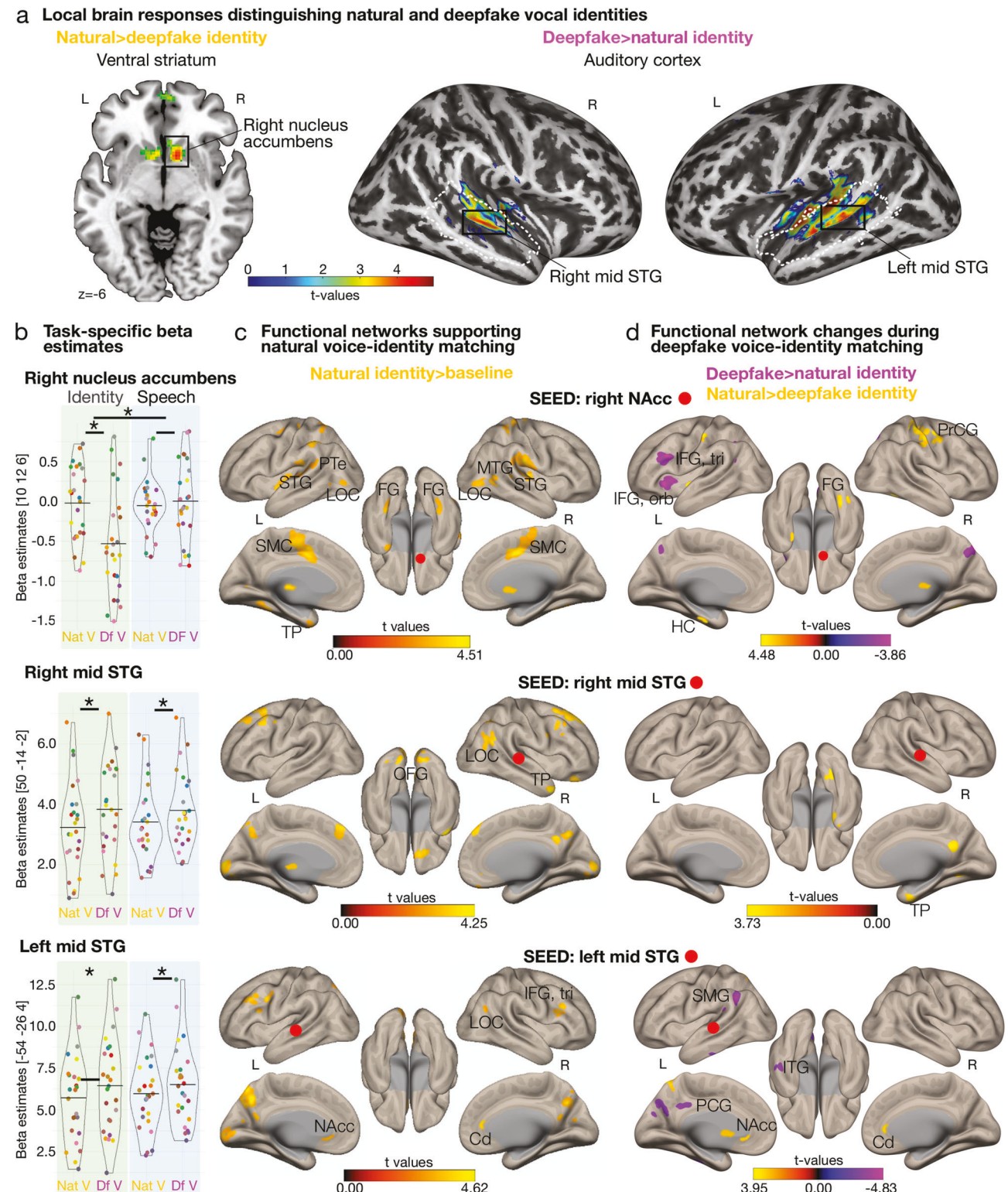

**a** Local brain responses distinguishing natural and deepfake vocal identities

**b** Task-specific beta estimates

**c** Functional networks supporting natural voice-identity matching

**d** Functional network changes during deepfake voice-identity matching

the sounds[27], which might trigger in-depth analysis both on a sound feature and on an object level[28,29].

**Ventral striatum coordinates a broad neural network for natural identity processing**

These distributed neural effects in the ventral striatum and the auditory cortex line up with the notion that person recognition is usually accomplished by a distributed brain network with specific functional connectivity patterns[30]. We therefore performed generalized psychophysical interaction (gPPI) analysis to determine functional networks with neural seeds from the central regions distinguishing natural and deepfake identity, that is right NAcc and bilateral AC (Fig. 2a). For all seed regions, first, we determined natural identity processing networks that contrasted the conditions for natural identity matching against neural baseline activity [$ID_{nat}$>baseline]. Second, we tested for network changes when targeted with deepfake identities by contrasting deepfake with natural identity matching [$ID_{df}$>$ID_{nat}$]

**Fig. 2 | Brain responses and functional networks for natural and deepfake speaker identities. a** Neural activity patterns for contrasting [$ID_{nat} > ID_{df}$]. White dashed line indicates the voice-sensitive regions evoked by the functional voice localizer scan. Second-level group t-maps, $p < 0.005$ corrected at the cluster level k > 47. **b** Beta estimates in right NAcc, right mid STG, and left mid STG for the conditions that were included in the interaction contrast ([$ID_{nat} > ID_{df}$] > [$SPEECH_{nat} > SPEECH_{df}$]). Plots show individual parameter estimates ($n = 25$) extracted from the maximum statistic for the contrasts shown in (**a**). Asterisks indicate significant effects of LMMs, *$p < .001$, We ran LMMs to specify the interaction effects. Circles indicate individual data and horizontal lines mean values. **c** Functional connectivity networks supporting natural identity matching (second-level group t-maps, gPPI, $p < 0.005$ corrected at the cluster level k > 47) from contrasting $ID_{nat} >$ baseline]; neural seeds as highlighted in (**a**). **d** Functional connectivity patterns with higher and lower connectivity for $ID_{nat} > ID_{df}$ and $ID_{df} > ID_{nat}$ (second-level group t-maps, gPPI, $p < 0.005$ corrected at the cluster level k > 47); neural seeds as highlighted in (**a**). Cd: caudate nucleus, FG: fusiform gyrus, HC: hippocampus, IFG, orb: inferior frontal gyrus, pars orbitalis, tri: pars triangularis, ITG: inferior temporal gyrus, L: left, LOC: lateral occipital cortex, MTG: middle temporal gyrus, NAcc: nucleus accumbens, OFG: occipital fusiform gyrus, PCG: posterior cingulate gyrus, PrCG: precentral gyrus, PT: planum temporale, R: right, SMC: supplementary motor cortex, SMG: supramarginal gyrus, STG: superior temporal gyrus, TP: temporal pole.

and vice versa [$ID_{nat} > ID_{df}$]. The gPPI analyses included both the identity and speech tasks to again control for basic sensory-acoustic effects.

During natural identity-matching conditions, the right NAcc was functionally connected with the pAC (planum temporale, PTe) and hAC (STG) implicated in voice signal analysis[28], the identity-encoding TP as a higher-order and multimodal association region[4], the visual cortex regions presumably involved in face representations[31], and the motor cortex (Fig. 2, Supplementary Table 7). In comparison to natural identity processing, during deepfake identity-matching processes, such neural connectivity for speaker-specific face and memory information seems to be unavailable given the atypical deepfake identities, and elevated recruitment of the inferior frontal gyrus (IFG) might be a strategy to balance the mismatching and thus the less expected deepfake voices[32]. The latter result was obtained when we directly compared neural networks for deepfake against natural identity matching ([$ID_{df} > ID_{nat}$], Fig. 2d, Supplementary Table 8).

## Auditory cortex connects with identity- and reward-coding nodes during natural identity processing
We next tested for networks that supported identity processing with neural seeds placed in the AC, both from a general perspective for natural identity processing (Fig. 2c) and when comparing the natural with the deepfake identity network (Fig. 2d). The natural identity-processing network revealed functional connectivity of the right AC, specifically with the identity-encoding TP and visual face-sensitive regions, whereas the left AC was mainly interconnected with the reward-predictive ventral striatum (NAcc) and the dorsal striatum (caudate nucleus, Cd) that is associated with reward-related actions[33] (Fig. 2c, Supplementary Table 7).

Unlike for the natural voice-identity network, we could only identify a limited deepfake voice-identity network centered on the left AC, with functional connections to inferior temporal cortex (ITG), supramarginal gyrus (SMG), and the postcentral gyrus as regions in the medial parietal cortex (Fig. 2d, Supplementary Table 8).

## Neural classification in ventral striatum is specific to natural identity information
Neural mechanisms for person-identity recognition are not only represented by univariate local brain activations and distributed connectivity networks but also by multivariate patterns of brain activations[34]. To further explore the fine-grained neural pattern representations of natural and deepfake identities, we performed a multivariate pattern analysis (MVPA) by using a support vector machine (SVM) learning approach. We ran a four-category classification scheme across our experimental conditions ($ID_{nat}$, $ID_{df}$, $SPEECH_{nat}$, $SPEECH_{df}$), focusing on regions of interest (ROIs) that we observed in our univariate activation (right NAcc, bilateral AC) and connectivity patterns (bilateral IFG pars triangularis, bilateral TP) that were central to the neural identity-processing network. The ROIs were based on anatomical definitions to avoid circular analysis with our functional brain data[35], and significance was tested for >25% accuracy above chance level (FFX one-sample t-tests[36]; corrected $p$-values for the number of subregions).

Consistent with our univariate analyses, we were able to predict neural patterns of natural vocal-identity processing ($ID_{nat}$) in all ROIs when the SVM classifier was trained on the neural patterns of the natural identity task (Fig. 3a–e, bottom left cell of confusion matrices, Supplementary Table 8).

When the SVM classifier was trained on neural patterns of the deepfake identity task ($ID_{df}$), deepfake identities were correctly decoded in the classic nodes of the voice-identity recognition network (AC, TP, and IFG), but not in the right NAcc (Fig. 3a–e, cell in line 3/column 2, Supplementary Table 8). Furthermore, when testing for task-specific effects, we observed that speech-related decoding ($SPEECH_{nat}$, $SPEECH_{df}$) was significantly above chance in all ROIs except for the NAcc (Fig. 3a–e, cell in line 2&1/column 3&4, Supplementary Table 9), adding further evidence for a very specific role of the right NAcc for processing natural voice identities.

In a cross-classification approach, we also tested whether the informational content of the neural pattern for deepfake identities ($ID_{df}$) can explain neural activity patterns for natural identities ($ID_{nat}$). A confusion analysis of neural patterns from the deepfake to natural identity task was indeed significantly above chance in all ROIs (Fig. 3a–e, cell in line 3/column 1, Supplementary Table 10). When trained on the neural information encoded in the natural identity task ($ID_{nat}$), neural patterns for the deepfake identity task ($ID_{df}$) could only be explained in the IFG (Fig. 3d, cell in line 4/column 2, Supplementary Table 10), suggesting naturalness-invariant encoding of speaker identity in the frontal cortex[37]. Altogether, across the larger neural identity network, the neural patterns for natural as compared with deepfake voice identities seem thus highly distinctive given the poor classifier prediction from natural to deepfake identities. In contrast, the neural information for deepfake as compared with natural voice identities seems broader and more unspecific given the high predictability of neural patterns from deepfake to natural identities.

## Social perception from voices is associated with auditory cortex activity
In the final step of the data analysis, we determined the social relevance of NAcc and AC activity that was central for distinguishing deepfake and natural identities. Participants rated deepfake voices significantly less natural, likable, and trustworthy than the natural voices (Supplementary Fig. 4, Supplementary Tables 11a and 12), which may be related to the decreased mean harmonicity in the deepfake voices[38,39]. On the neural level, we observed that neural activity differences in the AC but not in the NAcc were modulated by social ratings on vocal naturalness and likability (Fig. 4, Supplementary Table 13). The less natural and likable the deepfake voice was perceived compared with its natural counterpart, the greater the activity differences in the bilateral AC between deepfake and natural identity processing (Fig. 4a, b, Supplementary Table 13). No significant effects were observed for trustworthiness ratings on the neural responses in the NAcc and AC (Supplementary Table 13) nor for the speech task in the AC (Supplementary Table 14). The different brain-behavior regressions identified for each social dimension propose that each dimension, despite being correlated with each other (Supplementary Table 11b), encodes sufficiently distinct information.

## Discussion
This study tested the cognitive and neural abilities of humans to process audio deepfake information that is targeting a central feature of human beings, which is human identity. We recorded a set of natural speech utterances across several speaker identities and produced a complementary and matched set of deepfake utterances. The acoustic comparison of both

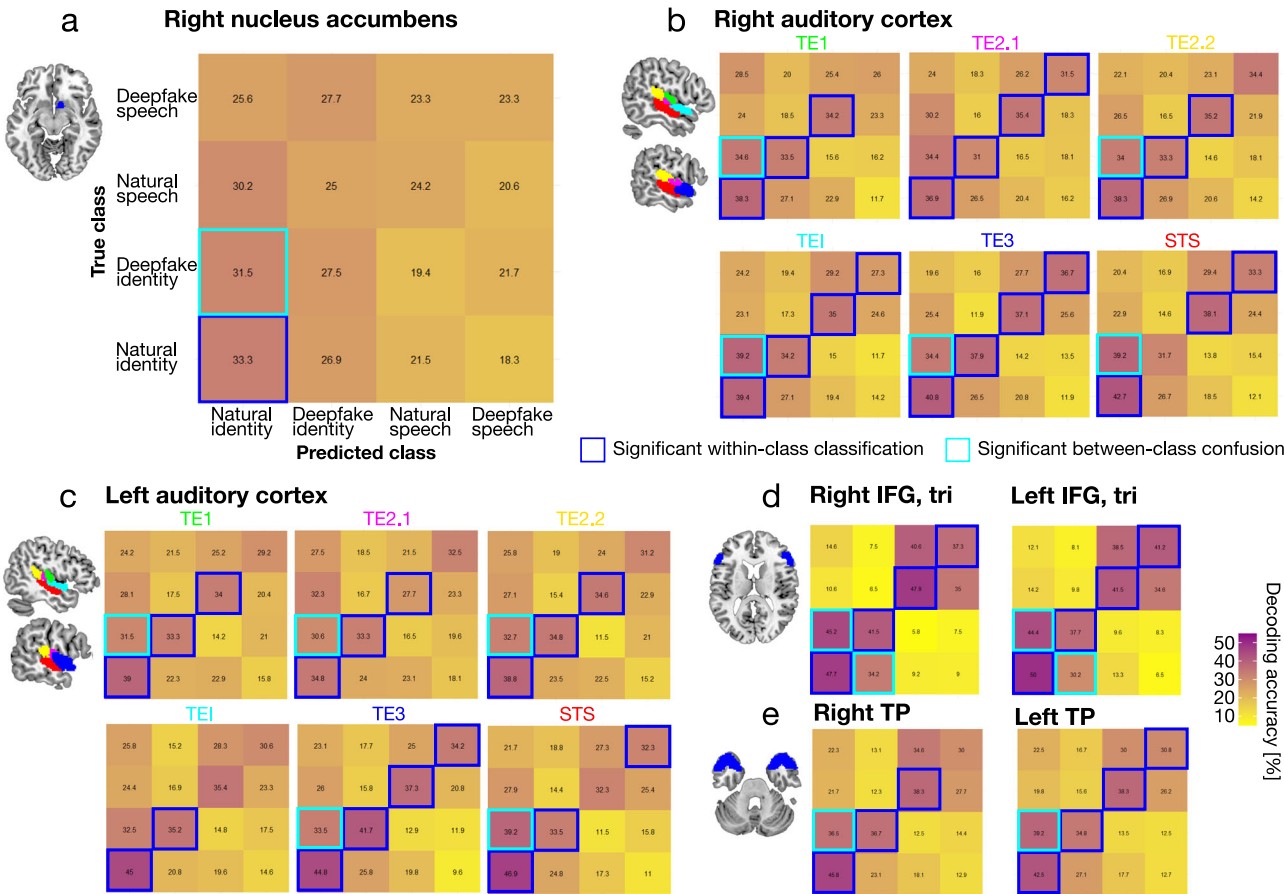

**Fig. 3 | Multivariate neural decoding of natural and deepfake speaker identities.** Multivariate decoding accuracies revealed by group-averaged confusion matrices comparing the frequency of the predicted sound class with the true sound class for **a** the right NAcc, **b,c** six subregions of the bilateral AC, **d** the IFG pars triangularis, and **e** TP. We anatomically defined region-of-interest (ROI) maps. Colored frames indicate decoding accuracies significantly above chance ($n = 24$, chance level 25%, one-sampled t-test, $p < 0.001$, Bonferroni corrected for the number of subregions per ROI).

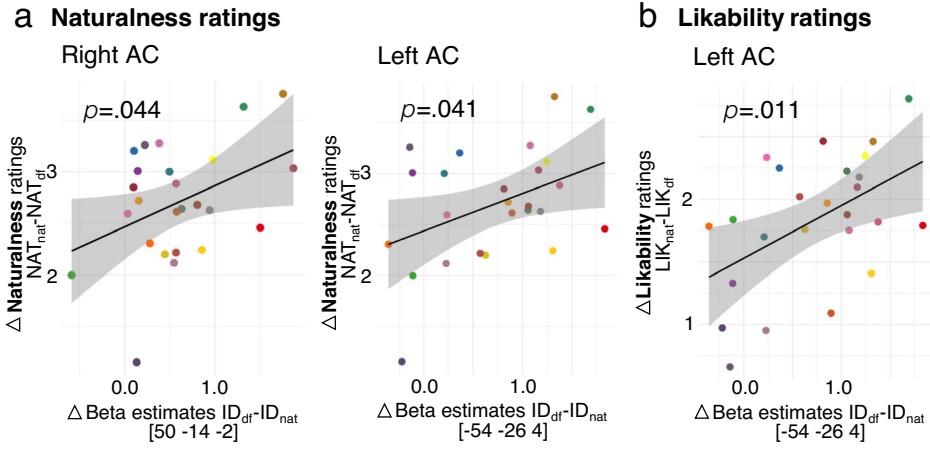

**Fig. 4 | Social perception encoded in AC activity.** Scatterplots show the association ($n = 25$, regression models, $p < 0.05$) between brain activity differences and measures of social perception. Circles indicate individual data and gray area shows +/- s.e.m. **a** Association of voice naturalness ratings with bilateral AC activity. AC activity for $[ID_{df} > ID_{nat}]$ increased for less natural rated deepfake voices ($NAT_{df}$) in relation to natural voices ($NAT_{nat}$). **b** Association of likability ratings with left AC activity. AC activity for $[ID_{df} > ID_{nat}]$ increased for less likable rated deepfake voices ($LIK_{df}$) in relation to natural voices ($LIK_{nat}$). For raw rating data see Supplementary Table 11A.

speech sets along a set of acoustic features mainly encoding indexical but also linguistic information[40] revealed commonalities along the mean and variation in fundamental frequency and variation in harmonicity. These commonalities highlight the notion that deepfakes voices can match natural voices to a certain degree, pointing to the quality of the voice conversion process. There were however also acoustic differences between natural and deepfakes voices, indicating that deepfake voice synthesis was on a high but not perfect level. Acoustic differences between natural and deepfake voice were found for the acoustic features of formant dispersion, voice jitter, mean

of vocal harmonicity, and speech rate. These features determine the perception of voice quality and rhythmicity, and their non-perfect resemblance in deepfake voice might lead to the impression of artificiality in deepfake voice and speech samples (Supplementary Fig. 4a). This perceived artificiality might however be a feature of deepfakes that should enable humans to detect deepfake identities to some degree.

With an identity matching task, we explicitly tested if deepfake voices could mislead humans to perceive deepfake voice identities as real (deepfake deception), or if humans would reject such deepfake voices as a non-match

to a natural voice identity (deepfake detection). Decision performance in the deepfake identity task was significantly lower as during the natural identity task. This decrease in performance was specific to the identity task since this performance difference was not found in the control speech task. The interaction observed between task and voice condition suggests that the decreased performance in identity matching of deepfake voice synthesis is not solely attributed to acoustic alterations introduced during synthesis. Rather, it also involves, to some extent, task-specific top-down effects. While the speech task and our acoustic analysis are useful to control for general acoustic effects on the identity task, they might not fully elucidate the distinct effects of specific acoustic features on each task. It is important to note that while deepfake identity matching was reduced, it was far from chance performance (i.e. around 70% correct responses), pointing to deepfake detection abilities in humans[12,13].

Next, we tested the neural mechanisms that humans recruit when faced with deepfake voice identities. By contrasting neural activity during the natural with the deepfake identity task condition, we found increased activity in the ventral striatum, which was located in the NAcc. The NAcc is central to the social brain, especially through its contribution to the (social) reward[41] and bonding circuits[42]. Concerning the processing of appealing sounds in humans, the NAcc is active, for example, when children listen to their mother's familiar voice[43] or when adults listen to pleasurable music[44]. The NAcc thus seems to encode the socially rewarding and appealing nature of favored stimuli with motivational relevance[45], potentially attributes incentive salience toward them, and motivates the approach to such hedonic cues[46]. Our set of natural voices was rated as highly likable and trustworthy (Supplementary Fig. 4b, c, Supplementary Tables 11a and 12), indicating their overall appeal and social relevance. More task-specifically, we speculate that the natural test voices during the natural identity task were expected by the listener (i.e. the internal identity representation is based on natural voices and more generally humans expect voices to be natural by default) and thus rewarding for the ongoing task as the incoming natural voice perfectly matches the internal identity representation that consequently elicited NAcc activity. In contrast to the natural identity task, NAcc activity was lower during the deepfake identity task, and this reduced NAcc activity might be a neural indicator of the resilience to deepfake identities. The reduced NAcc response may be related to two factors: first, the less social impact of our set of deepfake voices (Supplementary Fig. 4, Supplementary Tables 11a and 12) and, second, a confusion about the (non-)rewarding nature of the deepfake identity representation[47]. The deepfake voices closely but not perfectly resembled the acoustic features of natural identities, which may provoke unpredicted incorrect same/different identity decisions, and such uncertainties are associated with lower NAcc activity[48].

The similar pattern of task performance and the univariate NAcc response during the identity and speech task (Figs. 1d 2b) may suggest that the NAcc response simply reflects the level of task performance and the related reward/non-reward towards natural and deepfake voices irrespective of the task. However, multivariate findings suggest that the NAcc seems to be specifically sensitive to the identity information conveyed by natural voices. The NAcc correctly classifies only natural identity sounds but not natural and deepfake speech sounds, which are also associated with high task performance.

Further, we argue that it is unlikely that errors during the deepfake identity task were processed by the participants and thus modulated NAcc activation: first, participants received no feedback on their task performance, and second we observed no prolonged response time following erroneous trials[21] (i.e. no post error slowing effect, Supplementary Table 4).

When reversing the contrast to find higher brain activity during the deepfake compared with the natural identity task, we found significant activity in bilateral primary and higher-order AC. This activity was located within the general voice-sensitive cortex from posterior to anterior AC suggesting the contribution of the AC to different levels of voice decoding ranging from sensory-acoustic to higher-level processing[49]. The AC activity did not survive an interaction analysis and the AC classified both identity and speech sound classes with high accuracy. The decoding of voice

information in the AC is thus rather independent of the task, but seems to be crucial for deepfake sound processing. These auditory cortical resources may be a signal for elevated voice decoding to compensate the acoustically less informative and less naturally perceived deepfake signals (Fig. 4a), especially if these deepfakes were intended to mimic natural human features[50]. For our fMRI task, which was rather successfully performed for speech matching, these additional resources seem however insufficient to fully compensate for challenges in identity processing from deepfake voices (decreased identity-matching accuracy), but they might contribute to residual deepfake recognition abilities in humans (performance well above chance level). Again, one might ask whether the AC response linearly replicates behavioral performance in the identity matching task. However, the patterns of result differ not only between task performance and multivariate decoding results (i.e. decreased task performance for the deepfake identity condition vs. correct classification of all sound categories irrespective of task and sound condition), but also between task performance and univariate AC responses (i.e. significant task x sound condition interaction for task performance vs. non-significant task x sound condition interaction for univariate brain responses). Given the non-linear relationship between task performance and brain responses, we therefore argue again, as for the NAcc, that neural effects are likely modulated by more complex distinctions between the natural and deepfake vocal identities, and not just by task difficulty.

Together, the findings from the uni- and multivariate brain analyses suggest two functionally distinct brain systems that distinguish deepfake from natural vocal identity, with the AC more generally encoding perceptual deepfake sound information and the right NAcc processing specifically identity information of natural speakers being socially relevant and rewarding for the ongoing task.

Given the central relevance of the NAcc and AC in processing natural and deepfake voice identity, we also examined the broader functional network centered on these brain nodes. Functional connectivity with the NAcc as a seed region during natural identity processing was found with specific regions in the auditory and visual cortex as well as with the temporal pole (TP) region. This network likely supports the successful acoustic mapping from the incoming natural voice signal, in coactivation with the visual imagination or remembrance of a speaker's face information[51], to the internalized generative speaker model, established during the familiarization task, represented in the TP. These integral representations of speaker identity might be enhanced by memory information supplied by the hippocampus, as found in the specific network analysis when contrasting natural against the deepfake identity task. This might indicate that the NAcc generates a hedonic association with the informative and previously familiarized natural vocal identity signal. In comparison, during deepfake identity processing, this broader functional network centered on the NAcc was not present. This suggests that speaker-specific face and memory information is unavailable in this task condition, probably given the atypical and partly artificial representation of identity in deepfake voices. During deepfake voice identity processing, we only found elevated connectivity of the NAcc to the inferior frontal gyrus (IFG) that classified speaker identity independent of its naturalness. The IFG is typically active during challenging and explicit voice information decoding[52], which might be a neural mechanism to balance the mismatched identity information in deepfake voices[32].

Next to the NAcc, we also assessed the functional network centered on left and right AC, which showed activity for deepfake voice information processing. During natural identity processing compared to baseline, the right AC was connected to the identity-encoding TP and visual face-sensitive regions, and the left AC was connected with the reward-predictive ventral striatum (NAcc) and reward-coding dorsal striatum (Cd)[33]. The hemisphere-specific connectivity patterns suggest that the network centered on the right AC predominantly serves natural speaker-identity processing, whereas the network centered on the left AC encodes the social appeal of natural voice identities. A specific deepfake voice identity processing network was only found for the left AC. This network comprised connections

to a higher-order object processing region in the inferior temporal cortex (ITG) as an intermediate node, usually supplying information to the identity node in the TP[4], to the supramarginal gyrus (SMG) as an intermediate node of the dorsal auditory processing stream for dynamic sound-information processing[53] and sound-to-motor mapping[54], and to the medial parietal cortex (PCG) for identity information recollection[55]. Left AC might thus take over identity-related processes from the right AC during the more challenging deepfake identity task, but only up to an intermediate neural processing stage, which corresponds to the participants' intermediate ability to perceptually match deepfake-to-natural identities.

We want to clarify that our findings suggest neural sensitivity rather than selectivity to audio deepfake identities. It remains to be tested, whether the observed cortical-striatal brain network is similarly modulated by other voice modulations, whether natural or artificially induced by other techniques. Also, as audio deepfakes continue to improve their sound quality, it will be intriguing to observe whether their social acceptance also increases, and whether the neurocognitive system adjusts accordingly.

Taken together, our findings demonstrate the perceptual and neural level to which humans can be potentially deceived by instances of deepfakes that copy a central part of human beings: human identity. Copying human identity is a major target of many common deepfakes, and our data indeed showed that humans could be partly deceived by deepfake voice identities, taking the deepfake as an acceptable copy of the natural human instance. In some cases, humans can, however, also detect such deepfakes, which might be supported by a cortico-striatal brain network that decodes the socially rewarding nature of perceiving natural voice identities and the artificial nature of deepfake sounds. This cognitive and neural potential for deepfake detection in humans opens windows for empowering, sensitizing, and educating people to separate real from fake information in the digital age.

## Methods

### Participants
We invited 30 healthy volunteers to take part in the functional magnetic resonance imaging (fMRI) experiment (21 females; mean age 25.58 y, range=19–40 y). The participants reported normal or corrected-to-normal vision and we tested for normal hearing abilities via a pure-tone (250–8000 Hz) screening audiometer (Audiometer MA 25, MAICO, Berlin, Germany). No participant presented a neurological or psychiatric history, and participants performed within the neurotypical range on the Digit and Spatial Span tests, which are subtests of the WMS-R[56] that assess auditory and visual-spatial short-term working memory. All participants gave informed and written consent for their participation in accordance with the ethical and data security guidelines of the University of Zurich. The experiments were approved by the cantonal ethics committee of the Canton of Zurich. All ethical regulations relevant to human research participants were followed. Of the 30 participants, we included 25 in the final analyses (18 females; mean age 25.36 y, range=19–38 y). We had to exclude three participants because they did not meet the learning threshold of the speaker familiarization task. One participant had to be excluded because of excessive movement in the MRI scanner, and one participant fell asleep during the fMRI task. Because of a lack of comparable prior studies, we could not base our target sample size on an empirical effect size. We however used a typical sample size similar to recent fMRI studies of the human auditory system[57–61].

### Natural speaker recordings as stimulus material
We recorded stimuli from three German male speakers of standard German and two Swiss male speakers of native-like standard German (age range 19-34 y). One German speaker was a professional speaker and was used as the source speaker for the voice synthesis (see below). Speaker identities were all completely unfamiliar to the participants. We instructed all speakers to read sentences with a normal speech rate in an emotionally neutral intonation while they were videotaped. We audio-visually recorded 83 two-word declarative sentences, 35 five-word declarative sentences, and 3 five-word interrogative sentences. In addition, we audio-recorded 204 German

declarative and interrogative sentences of different lengths. This material was used for voice synthesis. Detailed technical information on the speech recordings can be found in the supplements.

### Synthesis of deepfake voice identities and speech utterances
The objective of voice synthesis is to copy speaker-specific paralinguistic voice features of a natural speaker identity (the target speaker) onto the linguistic speech material of another natural speaker (the source speaker). This enables the creation of completely new and deepfake speech material with the idiosyncratic voice profile of the natural target speaker who has never uttered the sentence before. In our study, the speech material of the professional speaker (degree in speech and language pedagogy) was the source for synthesizing our four target speakers. To synthesize deepfake voices, we used the open-source voice conversion (VC) software SPROCKET[16], which revealed the second-best sound quality scores for same-speaker pairs and the sixth-best quality for speaker similarity rating among 23 conversion systems submitted to the VC challenge in 2018[17] and requires relatively little training data (around 10 minutes of speech samples)[17]. SPROCKET implements a vocoder-free VC framework based on a differential Gaussian mixture model (GMM) and uses the same set of speech material for the source and target speaker (i.e. parallel data set) to model the conversion function. Detailed information on the synthesis can be found in the supplements.

### Acoustic analysis of natural and deepfake voices
For all natural and deepfake voices, we extracted a set of acoustic voice features reported to have behavioral relevance for person identity processing[19,62,63]. We used the standard settings of the Praat software[64] to extract the mean and standard variation of fundamental frequency, the mean and standard deviation of harmonicity, jitter, formant dispersion, and the ratio of the second and first syllable duration. To test whether acoustic features changed during the VC process, we ran linear mixed models (LMMs) implemented in the *lme4* R package (version 1.1-26[65]) in the R environment (Rstudio 1.3.1073). We first scaled all acoustic values with the *scale()* function to generate more comparable unit ranges across acoustic features. We ran separate models for each acoustic feature that was the respective dependent variable in each model. Models included sound condition as the fixed effect term and speaker and sentence as random intercepts. We considered effects significant if present at $p < 0.05$ and corrected for the number of acoustic feature models ($N = 7$) with the Bonferroni correction (seven models resulting in $p < 0.007$).

### General experimental procedure
For our full experimental set-up, we invited participants on two consecutive days. On day 1, participants first learned the vocal identity of four natural and unfamiliar male target speakers (referred to as the "speaker familiarization task"). After successful speaker familiarization (criteria described below), participants completed two fMRI experiments. In experiment 1 (referred to as the "FMRI matching task"), we tested speaker identity and speech matching when presented with the natural and deepfake synthetic voices of the previously familiarized speaker identities. The identity-matching task closely resembles everyday social tasks, such as tracking speaker identity in a multispeaker conversation. In experiment 2 (referred to as the "functional voice localizer"), we mapped voice-sensitive brain areas of the auditory cortex (AC)[28,66,67]. Testing on day 1 lasted 2.5 hours. On day 2, participants rated the naturalness, likability, and trustworthiness of the natural and deepfake voices presented in the fMRI matching task. Further, we tested short-term memory abilities (digit and block span test of the WMS-R) and hearing thresholds. Complete testing on day 2 lasted 1 hour. Stimuli of all tests, except the functional voice localizer, were presented and responses were recorded with Presentation software (version 21.1, Neurobehavioral System Inc., CA, USA). The functional voice localizer was implemented in Cogent (Matlab, version 2016b, The MathWorks, Inc., MA, USA).

### Speaker familiarization task

Participants first learned and recognized the four male target speakers presented in the main fMRI task. The natural target speakers' voices were learned together with the moving speaker's face and name. We chose a multimodal familiarization approach to make speaker familiarization as natural and effective as possible. The speaker-familiarization task contained seven learning and seven testing phases in alternating order. The first two learning and testing phases were repeated after the fifth test. During learning, participants were presented with audio-visual videos of the four speakers uttering different five-word German declarative sentences. In parallel, the written name of the speaker was presented on the screen. Participants were asked to learn the respective voice-face-name associations. The number of sentences varied across learning phases: first phase two blocks and four sentences per speaker; second phase two blocks and three sentences per speaker; third, fourth, and fifth phase two blocks and two sentences per speaker.

During each testing phase, participants listened to five different auditory-only sentences per speaker. After each sound presentation, participants performed a four-alternative forced-choice task in which they selected the face/name matching the speaker's voice. To avoid prosody-driven identity recognition, we presented different types of sentences during the third (five-word interrogatives) and fourth (two-word declaratives) testing phase. Response feedback was presented in all testing phases except the fifth. Green crosses indicated correct and red crosses incorrect responses. During the first and second testing phase, the correct combination of the audio-visual video and the name was repeated, irrespective of correct or incorrect response. During each learning and testing phase, the order of the speaker and sentence presentation were randomized. The complete task took 25 min. Only participants who performed above 80% correct in the last two testing phases completed the rest of the study. With that, we ensured that only participants who established a robust representation of the natural speaker identity participated in the fMRI speaker-speech task. All stimuli were presented via headphones (Sennheiser HD 201, Wennebostel, Germany) and a laptop (Latitude 4300, Dell, Round Rock, TX, USA).

### FMRI matching task on speaker identity and speech content

In preparation for the fMRI matching task, participants passively listened to one block of sentences per speaker identity, spoken either by the natural or the deepfake voice. This step aimed to reduce any potential perceptual "surprise" effect during the main fMRI task, especially in the deepfake task conditions. A block included four different sentences, with sentence repetitions across voice-condition blocks. The order of blocks was randomized across participants. Along with the voice, we displayed the picture of the speaker's face and the name on the screen. Importantly, we did not instruct participants that they would listen to deepfake voices of the natural speakers, instead we stated that they would hear voices that might sound slightly different from those of the natural speakers. Also, we trained participants on the exact fMRI task procedure. To familiarize participants with the fMRI task procedure, we had them complete one practice block per task condition: natural identity (ID$_{nat}$) task, deepfake identity (ID$_{df}$) task, natural speech (SPEECH$_{nat}$) task, deepfake speech (SPEECH$_{df}$) task. All stimuli were presented via headphones (Sennheiser HD 201, Wennebostel, Germany) and a laptop (Latitude 4300, Dell, Round Rock, TX, USA) outside the MRI room.

The fMRI matching task included two task conditions (identity and speech task), each with two sound conditions (natural and deepfake voices) (Fig. 1b–c). During the entire fMRI matching task, participants exclusively listened to sounds, no face pictures or names were presented. Stimuli were organized in blocks with one sentence presented at the beginning of a block (target trial, 1.5 s with 1 s auditory presentation), followed by a sequence of 12 sentences, these being the test trials (1.5 s with 1 s auditory presentation). The sentences were spoken by four different speakers, and within each block there were four phonologically similar sentences (e.g., "Er fällt" (English: "He falls"), "Er fehlt" (English: "He misses"), "Er föhnt" (English: "He blow-dries"), "Er fühlt" (English: "He feels")), ensuring a comparable level of task

difficulty between the identity and speech task. At the beginning of each task block, a written screen instructed the participants to perform either the identity or the speech task (1.5 s). Participants, however, were naïve as to whether they would listen to natural or deepfake voices. In the identity task, participants were instructed to memorize the speaker identity of the target trial and to decide after each of the test trials whether the speaker identity of the test sentence matched the target speaker identity, irrespective of the linguistic content (Fig. 1c). In the speech task, participants listened to the same stimuli, but were instructed to memorize the linguistic content of the target sentence and decide whether the linguistic content of the test sentence matched the content of the target sentence, irrespective of the speaker identity (Fig. 1c). Of the 12 test trials, 4-6 contained the target information (identity, speech). In the natural voice condition, target and test sentences were uttered by natural voices. In the deepfake condition, participants matched deepfake test utterances to a natural target utterance. The same set of sentences was presented in the natural and deepfake identity and speech tasks.

Each task block was 21 s long and was followed by a silence block (18 s) during which a fixation cross was presented on the screen. During the fMRI matching task, we presented 20 blocks per condition in a pseudo-random order with a maximum of two consecutive blocks of the same task condition (80 blocks in total). Participants indicated their same/different responses with the right hand via two buttons on a button box. Half of the participants were instructed to press the left button with the index finger when the test and target trial matched and the right button with the middle finger when the test and target trial differed. The other half of the participants indicated match responses with the middle finger and non-match responses with the index finger. The experiment was completed in four runs, with short breaks in between. One run lasted 14 min and the fMRI matching task took 56 min in total. After the fMRI data acquisition, a written debriefing was conducted whereby participants reported on their ability to concentrate during the identity and speech task and on potential strategies used during the tasks.

### Functional voice localizer scan

We presented recordings of 70 voice sounds (speech, nonverbal/non-speech) and 70 non-voice sounds (animal, natural, and artificial sounds) as stimuli during a functional voice localizer scan[28,67]. Stimuli were presented in an event-related design, with 10% of the sounds repeated randomly for a 1-back task, in which participants were asked to press a button on the consecutive repetition of a sound. Each sound lasted 500 ms. All sounds were presented one time in random order, with a jittered ITI of 4.0-5.5 s and a sound intensity level of 68 dB SPL. We included the functional voice localizer task to test whether activations observed during the fMRI matching task overlapped with the classic voice-sensitive regions[67].

### FMRI data acquisition

Structural and functional images were recorded on a 3T-Philips Ingenia with a standard 32-channel head coil. High-resolution structural images were acquired by using T1-weighted scans (field of view, 250 × 250 × 180.6 mm; matrix, 256 × 251; 301, 1.20 mm overlapping sagittal slices). Functional images were recorded by using a T2*-weighted echo-planar pulse (EPI) sequence (TR 1.6 s, TE 30 ms, flip angle 82°; in-plane resolution 220 × 114.2 mm, voxel size 2.75 × 2.75 × 3.5 mm; gap 0.6 mm, ascending acquisition and in anterior commissure (AC) – posterior commissure (PC) orientation) covering the whole neocortex. In the fMRI task, 242 volumes were acquired in each run for each participant (total 968 volumes). In the functional voice localizer experiments, 407 volumes were acquired for each participant in one run. In both fMRI experiments, volumes were acquired continuously with a TR of 1.6 s.

### Social perception ratings

After completing the fMRI experiments, participants listened again to natural and deepfake voices presented in the fMRI matching tasks and were asked to rate the perceived sounds' naturalness, likability, and trustworthiness on a five-point Likert scale. The naturalness and trustworthiness

scale ranged from 1 to 5 with the following labels: 1 "strongly disagree," 2 "disagree," 3 "neutral," 4 "agree," 5 "strongly agree." The likability scale ranged from −2 to 2 with the following labels: −2 "very unpleasant," −1 "unpleasant," 0 "neutral," 1 "pleasant," 2 "very pleasant." Sounds were presented in randomized order and after each presentation, participants first rated the naturalness, followed by the likability rating, and finally the perceived trustworthiness. Before each rating, sound could be replayed once, according to the participant's need. To make ratings more comparable across participants, especially for the likability and trustworthiness ratings, we gave short written instructions. For the likability ratings, we wrote, "How likable do you perceive the voice, or would you like to listen to the voice for longer?"; for the trust ratings, we wrote, "How trusting do you perceive the voice, or would you trust this speaker to keep a secret?"; and for the naturalness ratings, we wrote, "How natural do you perceive the voice?" We instructed participants to respond as quickly as possible. The complete task took approximately 60 min and was separated into two runs.

## Behavioral data analysis

**Identity and speech matching task.** We ran LMMs to statistically test interactions and main effects of the fMRI matching task. For LMMs we used the *lme4* package (version 1.1-26,[49]). We conducted the model reduction approach to find the best fitting model structure for both random and fixed effects. We started with the full model including task-based percentage of correct scores per participant and the task and sound condition and the interaction of both as fixed effects. As random effects we included participant with random slope for sound condition and task. The best fitting model included the full structure of fixed effects and a reduced structure of random effects:

$$lmer(percent\ correct \sim task * sound\ condition + (1|participant))$$

Post hoc tests were conducted by using *emtrends* from the *emmeans* package (version 1.5.5-1).

**Post-error slowing (PES).** We quantified post-error behavior by comparing mean reaction times on trials following error trials with the trials directly preceding the same error. Only errors that were both preceded and followed by correct trials were considered, following the ´robust´ PES approach[20]. We ran linear models with the R package *stats* (version 4.0.3[68]) to statistically compare PES between task and sound condition:

$$lm(PES \sim task * sound\ condition)$$

In addition, we directly tested whether reaction time differences around erroneous trials during the deepfake identity task were greater than zero, i.e. for a PES effect during the most challenging task.

**Social voice perception ratings.** To compare the ordinal rating outcomes for the natural and deepfake voices, we ran Cumulative Link Mixed Models (CLMM) using the "ordinal" package (version 2020.8-22[69]). We used the model reduction approach to get the best fitting model structure of fixed and random effects. The full model included the ordinal rating values as dependent variable and sound condition as fixed effect. As random effects we added participant and speaker with random slopes for sound condition. After model reduction, the full model fitted the data best. We ran separate models for each rating dimension:

$$clmm(rating\ value \sim sound\ condition + (1 + sound\ condition|participant)$$
$$+ (1 + sound\ condition|speaker))$$

For all behavioral analyses, we considered effects statistically significant if the 95% confidence interval did not contain 0 referring to $p < .05$.

## Functional brain data preprocessing

For motion correction, the functional data were first manually realigned to the AC-PC axis. Following SPM realignment, we manually inspected

individual motion parameters, ensuring none exceeded 2.75 mm, corresponding to the sampled voxel size. Each participant's structural image was co-registered to the mean functional image and then segmented to allow estimation of normalization parameters. Using the resulting parameters, we spatially normalized the anatomical and functional images to the Montreal Neurological Institute (MNI) stereotactic space. The functional images for the main experiment were resampled into 2mm³ voxels. All functional images were spatially smoothed with a 6 mm full-width half-maximum isotropic Gaussian kernel.

## Statistical analysis of functional brain data

**FMRI matching tasks.** For the first-level analysis, we used a general linear model (GLM), and all blocks were modeled with a stick function aligned to the onset of each task block, which was then convolved with a standard hemodynamic response function. We separately modeled the four task blocks: $ID_{nat}$, $ID_{df}$, $SPEECH_{nat}$, and $SPEECH_{df}$ task. Thus, the GLM included four regressors. We also included six motion correction parameters as regressors of no interest to account for signal changes not related to the conditions of interest. Contrast images for each condition were then taken to a random-effects, group-level analysis to investigate the neural processing for natural and synthetic sounds during the identity and speech task. Consistent with previous studies reporting voice-sensitive response in the bilateral temporal lobe[29,49,66,70] integrated into a broader brain network[52,66,71,72], we conducted the contrast analysis at the level of the whole brain.

**Functional voice localizer.** For each participant, we modeled 70 vocalizations (speech and non-speech) as one regressor, and 70 non-vocal sounds (artificial, animal, and natural sounds) as a second regressor. We included as regressors-of-no-interest the 14 sound repetitions (10%) from a one-back task. The fMRI model included six additional regressors of head motion that were estimated in the realignment step during pre-processing. At the second level, we computed the linear contrast of the first two regressors, [vocal minus non-vocal], served to identify voxels that are more sensitive to vocal than to non-vocal sounds.

All group results were thresholded at a combined voxel threshold of $p < 0.005$ corrected for multiple comparisons at a cluster level of k = 47. This combined voxel and cluster threshold corresponds to $p = 0.05$ corrected at the cluster level and was determined by the 3DClustSim algorithm implemented in AFNI software (afni.nimh.nih.gov/afni; version AFNI_18.3.01), including the recent extension to estimate the (spatial) autocorrelation function (ACF) according to the median estimated smoothness of the residual images. The cluster extent threshold of k = 47 was the maximum value for the minimum cluster size across contrasts of the fMRI matching task and the functional voice localizer scans.

## Brain regions-of-interest (ROIs) analysis

Naturalness-sensitive regions-of-interest (ROIs) for the connectivity and brain-behavioral analysis were the right nucleus accumbens (NAcc), the right mid superior temporal gyrus (STG), and the left mid STG. We functionally localized the right NAcc with the univariate contrast '$ID_{nat}>ID_{df}$' and the right mid STG and the left mid STG with the contrast '$ID_{df}>ID_{nat}$' all at $p < 0.005$, k > 47. Final ROIs were defined as spheres positioned around the group peak coordinate of the right NAcc (x = 10, y = 12, z = -6, 3 mm radius for the subcortical region), the right mid STG (x = 50, y = -16, z = -4), and the left mid STG (x = -42, y = -26, z = 2, 5 mm radius for the cortical regions).

For the multivariate voxel pattern analysis, we defined ROIs based on probabilistic anatomical maps to avoid circular analysis[35]. We used neuroanatomical atlases to define the right NAcc[73] and six subregions of the AC covering sensory primary up to higher level secondary AC regions, including primary AC with Heschl's gyrus Te1, secondary AC with STG including planum temporale Te2.1&Te2.2, temporo-insular region with planum polare TeI, higher AC with anterior/mid STG Te3, and superior temporal sulcus STS1[74]. In addition, we included the bilateral inferior frontal

gyrus (IFG) pars triangularis (IFG45)[75] and temporal pole (TP)[76] as ROIs, both key nodes of the voice network[4,30] and functionally connected with the NAcc and AC during the identity-matching task. We thresholded the NAcc, AC, and TP maps by 20% and the IFG maps by 30% to restrict maps to anatomically meaningful structures. All maps were registered with a functional EPI image.

### Task-based functional connectivity analysis of brain data

For the functional connectivity analyses we followed the standard procedure as detailed in the CONN-Toolbox manual. We computed seed-to-voxel analysis with the naturalness-sensitive ROIs: right NAcc, right mid STG, and left mid STG. Using the import function for SPM files, we automatically imported design specific information (runs, task conditions) and the pre-processed functional data (including movement parameters) into the CONN-Toolbox. In addition, we imported pre-processed subject-specific anatomical images and the functionally predefined ROI maps. Spurious sources of noise were estimated and removed by using the automated toolbox preprocessing algorithm, and the residual BOLD time series was band-pass filtered by using a low-frequency window ($0.008 < f < 0.09$ Hz) to focus on slow-frequency fluctuations while minimizing the influence of physiological, head-motion, and other noise sources. We performed a generalized psycho-physiological interaction (gPPI) analysis, computing the interaction between the seed BOLD time series and a condition-specific interaction factor when predicting each voxel BOLD time series. In contrast to standard PPI, gPPI allows the inclusion of the interaction factor of all task conditions simultaneously in the estimation model to better account for between-condition overlaps. For our analysis, we included all four task conditions in the gPPI model, allowing relative specific interpretations on the identity task condition while controlling for overlapping effects introduced by the speech task. Seed-to-voxel connectivity maps are based on univariate regressions. For the second-level analysis, we specified a baseline contrast $ID_{nat} >$ baseline and between-condition contrasts $ID_{nat} > ID_{df}$ and $ID_{df} > ID_{nat}$. We considered effects to be significant, in line with the univariate analysis, if present at $p < .005$ uncorrected with a cluster threshold of $k > 47$.

### Multivariate voxel pattern analysis (MVPA) of functional brain data

Multivariate decoding was done with the linear support vector machine classification ($C = 1$). For the MVPA, we fitted each task block of the fMRI matching task with a canonical hemodynamic response function, yielding 20 beta images per task condition and a total of 80 beta images per participant. We ran separate ROI decoding in the right NAcc, subregions of the bilateral AC, bilateral IFG, pars triangularis, and bilateral TP. Our goal was to investigate the classification accuracy within the class of natural and deepfake voice identity and the confusion between natural and deepfake identities in the respective ROIs. For that, we submitted the beta images of all four task conditions to a leave-one-run-out cross-validation procedure by using a linear support vector machine classification approach ($ID_{nat}$, $ID_{df}$, $SPEECH_{nat}$, $SPEECH_{df}$ chance Level=25%). Training the classifier on both the identity and speech categories allowed more identity-specific interpretations of the results. The output was specified to reveal a confusion matrix comparing the frequency of the predicted label with the true label. This procedure was repeated for all participants for each ROI, and the resulting confusion matrices were averaged across participants. We statistically tested whether accuracies and confusions were significantly above chance level (25%) with fixed-effects one-sample t-tests[36], and we corrected for multiple decoding models within the AC subregions ($N = 6$) with Bonferroni correction. We had to exclude one participant from the analysis because the number of beta images per task condition was unbalanced, resulting in 24 participants in the MVPA.

### Brain-behavior association analysis

For testing the relation between social perception and neural responses in the brain regions showing differential activity to natural and deepfake voice identities, we performed the following steps. First, we calculated the difference in naturalness, likability, and trustworthiness ratings: natural voice ratings minus deepfake voice ratings. Second, we calculated difference scores for the beta estimates of the natural and deepfake identity task: parameter estimates $ID_{nat}$–parameter estimates $ID_{df}$. We extracted beta estimates from the NAcc, right AC, and left AC ROIs by using the toolbox MarsBaR[77]. We ran linear regression models implemented in the *stats* R package to correlate the beta estimate difference scores to the social perception scores. Brain responses were entered as the dependent variable and the social perception scores as the independent variable. For each ROI, we ran separate models for each behavioral difference score. We considered effects to be statistically significant if the 95% confidence interval did not contain 0 referring to $p < 0.05$. Given the task-general neural effect in the AC, we ran the same analyses for the right and left AC on the beta estimates of the natural and deepfake speech task: parameter estimates $SPEECH_{nat}$ – parameter estimates $SPEECH_{df}$.

### Statistics and reproducibility

For all analyses, we report detailed statistical approaches in each of the relevant sections above (i.e. acoustic analysis, behavioral data analysis, univariate, functional connectivity, and multivariate brain response analyses (whole brain and ROI analyses) as well as correlations between univariate brain responses and social perception ratings).

Preprocessing and univariate statistical analyses of functional images were performed with the Statistical Parametric Mapping software 12 on MATLAB R2018b. The task-based connectivity analysis was carried out in the matlab-based Functional Connectivity Toolbox (CONN toolbox)[78] and the Multivariate Pattern Analysis in The Decoding Toolbox (TDT version 3.999[79]). All behavioral analyses were done in the R environment (Rstudio 1.3.1073). Functional ROIs were generated with the matlab-based MarsBaR toolbox[80]. Probabilistic maps of the AC subregions were downloaded here: TeI: https://search.kg.ebrains.eu/instances/Dataset/35f2107d-2e3c-41ae-b26a-83b101829346; Te1: https://search.kg.ebrains.eu/instances/Dataset/8a105954-a724-428d-aed8-6c8d50fe4218; Te2.1: https://search.kg.ebrains.eu/instances/Dataset/a5589ffe-5148-4006-8d8d-8bcf411f750b; Te2.2: https://search.kg.ebrains.eu/instances/Dataset/a8247117-3349-49b7-a2f4-aa62b5fdd115; Te3: https://search.kg.ebrains.eu/instances/Dataset/03b1cb7b-f142-4612-bbd3-10fc7743bf13; STS: https://search.kg.ebrains.eu/instances/Dataset/61460dd7-7696-485e-b638-407e7b7dc99f. Probabilistic maps of the bilateral NAcc were downloaded from https://neurovault.org/collections/3145/, and of the bilateral IFG45 here: https://search.kg.ebrains.eu/instances/Dataset/1db64761-aad6-4d3d-b67c-2a9ef4df6e47. Probabilistic maps of the bilateral temporal pole were extracted from the Harvard-Oxford cortical and subcortical structural atlases in FSL. All probabilistic maps were converted to SPM-readable and co-registered ROIs using MarsBar.

The flat representation of the human cortex displayed in Fig. 2a was created with the CAT12.5 toolbox and results of the connectivity analysis displayed in Fig. 2c, d were generated with the CONN toolbox.

### Data availability

The experimental data that support the findings of the study, including the source data for the graphs, and the auditory material are available from https://doi.org/10.17605/OSF.IO/89M2S.

### Code availability

The code that supports the findings of this study is available from https://doi.org/10.17605/OSF.IO/89M2S.

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

## Acknowledgements

This study was supported by the Swiss National Science Foundation (SNSF 10DL15_183152 to SF/VD; SNSF PP00P1_157409/1 and PP00P1_183711/1 to SF).

## Author contributions

C.R. contributed to design conceptualization, investigation, methodology, data analysis, visualization, and writing of the original draft. T.K. contributed to the voice conversion methodology. E.P. contributed to the investigation. V.D. contributed to funding acquisition, design conceptualization, and methodology. S.F. contributed to funding acquisition, design conceptualization, methodology, data analysis, supervision, and writing. All authors provided critical revision of the manuscript and reviewed and approved the final manuscript.

## Competing interests

The authors declare no competing interests.
