## [Peer review file · Communications Biology]

Reviewers' comments:

Reviewer #1 (Remarks to the Author):

This paper reports the responses of the brain to natural voices and deepfakes created by a particular software (sprocket). The fMRI study uses an elegant “von Kriegstein”-design whereby participants have to complete an identity and speech matching task on the same stimuli (natural and “unnatural”/deepfake voices). Participants identity matching performance was worse for the unnatural voices than the natural ones. Results of a variety of different analyses of the MRI study showed that natural and unnatural voices are processed similarly (but not identically).

The paper is well written and appears to be thoroughly analysed but I do not recommend publication for the following reasons:

- (1) Given the unnaturalness of the deepfake voices the results are as one would expect. Performance drops for the unnatural voices. The brain, to an extent, picks up on that “unnaturalness” (or strangeness). The identity and speech task are not matched for difficulty (speech task easier) and so performance on the identity task is affected but the performance on the speech task is not.
- (2) In the identity matching task, natural voices elicited activity in the nucleus accumbens which the authors interpret to mean that the brain prefers natural voices because they may be more socially rewarding. Note that this pattern of activation was not found for the speech matching task (p7) which in my opinion overthrows their interpretation. The role of the basal ganglia in auditory and, more specifically, language perception and learning, is relatively well studied (e.g. Krishnan et al., 2016 TICS). The author’s interpretation re: the role of the NAcc in their experiment is post-hoc. There is nothing in the data to confirm that what is going on is indeed related to “reward processing” regarding the natural vs “fake” voices in one task but not another.
- (3) There are no hypotheses guiding the work and the work is not introduced or discussed in light of a fairly large literature on voice identity perception, including voice morphing (which is not unlike the authors’ current manipulation).
- (4) The findings are likely specific to that particular voice conversion manipulation (sprocket) the authors decided to use. Sprocket seems to dramatically decrease the harmonicity of the manufactured voices. Decreased harmonicity has previously been linked to decreased likability, trustworthiness etc (e.g. see Baus et al. (2019)).
- (5) I suspect that ratings of naturalness, likability and trustworthiness were correlated. Is that the case? If so, how can you differentiate between any of these variables in your regression analyses (will also affect results illustrated in Fig 4).
- (6) The paper misses some basic details re: scan acquisition – what scanner was used? What tesla? How many channels has the headcoil? The voxels were resampled to 2mm³ but what was the original acquisition (in the attached “reporting summary” the reported voxel size is unusually large – 2.75x2.75x3.5mm with .6mm gap but the pdf is difficult to read due to overlapping text so I may have misread)? Do the authors have the spatial resolution to draw strong conclusions about the neuroanatomical substrates of their reported activations?

Reviewer #2 (Remarks to the Author):

Summary:

In an fMRI study (n=25), the authors claim to show two functionally distinct brain systems that distinguish deepfake from natural voice identity. The auditory cortex (AC) is suggested to more generally encode perceptual deepfake sound information (and other sound information) and the right Nucleus accumbens (NAcc) processes specifically identity information of natural speakers being socially relevant and rewarding for an ongoing task. Auditory cortex (parts of the general voice-sensitive cortex) showed increased activity for deepfake compared to natural voice and allowed for the decoding of the vocal acoustic pattern and deep fake level. NAcc activity was suggested to function as “deepfake detector” related to the preference natural speaker identities (i.e. reduced NAcc activity was observed to deepfake vs. natural voices, and activity correlated with likeability ratings; NAcc allowed to classify natural, but not deepfake from natural voices). Furthermore, connectivity analyses (generalized psychophysical interactions, gPPI) were performed with the right NAcc and bilateral ACC as seeds to investigate identity networks for natural voices (test against baseline) and deepfake detection (condition comparison). For natural voices and right NAcc seed the network comprised: pAC (planum temporale), hAC (STG) and TP (multi-modal association cortex), visual cortex and motor cortex (IFG). The deepfake networks showed no visual cortex recruitment and increased IFG recruitment. For the right AC seed the natural identity network comprised the TP and face sensitive regions, the left AC seed the NAcc and the dorsal striatum. For the deep fake identity comparison only a network for the left AC with ITG and SMG and PC were found. Furthermore, multivariate pattern analysis (MVPA) with SVM learning was performed classifying the speaker identity and the speech content for natural and deepfake conditions focusing on ROIs (right NAcc, bilateral AC, bilateral IFG, bilateral TP). Other than natural voices deepfake voices were only decoded from AC, IFG and TP not NAcc. The study used voice identity clones generated with deepfake technologies (SPROCKET; with deepfake versions of the natural voice of 4 speakers). First, in an acoustic analysis seven features that contribute to voice identity were compared across natural and artificial voices. No differences in mean and variation of fundamental frequency (relevant for vocal pitch perception) or vocal harmonicity (relevant for voice quality) were observed. However, there were significant differences in less central features of voice identity (related to vocal timbre, micro-fluctuations of vocal pitch and vocal sonority and speech rate). Second, participants were familiarized with the natural voices in a task where faces, names and voices were presented. During the fMRI an identity matching task was conducted (does the test voice match the target voice irrespective of content; the target voice was always natural, in two conditions the test voice was either natural or deepfake). The performance for the matching of natural voices was on average 80%, for matching deepfake to natural voices 68.94%. The authors suggest that the reduced ability to match natural and deepfake voices is due to the not perfect “cloning of voice features” by the algorithm. In a control task, participants performed a speech content matching task on the identical voices, where no differences in behavioral performance and in neural activation between natural voices and deepfake were observed. Participants rated deepfake voices significantly less natural, likable (and trustworthy) than the natural voices, this difference was correlated with neural activity

Comments:

This study investigates a timely question about our ability and neural basis of distinguishing natural and deepfake voice identities, which may inspire new thoughts in the field. Furthermore, it provides relevant insights into voice-specific neural processing in more general. It includes a

comparison of the acoustics of natural and deepfake voices, which is interesting itself. The study is very well written and the analysis seem advanced/state of the art. A control condition and analyses are included. I have several comments that aim to provide thoughts for improvement and some methodological clarification questions. I think the study would benefit from more clearly stating the conceptualization of the basis for (neural) deepfake detection. And whether such processes are thought to be specific to deepfake detection. If it was based on social relevance, I would assume this mechanism is not only involved in deepfake voice processing but is a gradual process also involved in the evaluation of natural voice familiarity. Furthermore, such a detector would fail when deepfake voices gain social relevance. I am wondering whether this interpretation is valid in this study, as participants were only familiarized with natural but not with deepfake voices which might have impacted such processes (if I understood correctly). Could a “neural deepfake detector” instead be based on the physiological plausibility of voices? This seems to me a more genuine difference between natural and deepfake voices. Second, about the control task: The speech content task suggests that both stimuli were well perceivable, however, beyond that differences in voice processing relevant acoustics might not affect speech content processing to the same degree (e.g., they might be in the focus of attention for voice processing). Is it possible to isolate deepfake voices that were acoustically more strongly overlapping with the natural ones and compare them to those that were not? (I assume that not enough voices were tested for this comparison). Alternatively, the appropriateness and limitations of the control task might be discussed more clearly.

Detailed comments:

- About the familiarization task: this sounds like ppl were familiarized only with the natural voices. Is it possible that the neural differences reflect familiarity to a certain extend?
- P. 7 “For the functional brain data, we [...]” In the previous paragraph several regions of interest are mentioned, here it sounds like a whole brain analysis was performed. Could you clarify?
- P. 8 “In comparison, during deep-fake identity matching [...]” This sounds to me like it describes the networks recruited during the deepfake voice processing. However, in case I understood it correctly, the network analysis was a comparison of natural and deepfake voice processing and not a contrast with baseline (as for natural voices), I would revise this to make clear it’s not the network “per se” but the difference.
- P. 8 “[...] speaker-specific face and memory information seems to be unavailable given the atypical deepfake identities, [...]” If I understood correctly and only the natural voices were paired with faces and names in the familiarization period, isn’t that a problem for this interpretation? This might not be a general characteristic of deepfake processing but rather of the learning in this paradigm?
- P. 10 “[...] suggesting modality-invariant encoding of speaker identity in the frontal cortex [...]” Could you clarify why the cross-condition decoding suggests modality independent processing in IFG? Is “modality” referring to deepfake vs. natural?
- Was there any brain activity that was related to the task performance (whether participants were deceived or not by the deepfake?). This might be interesting in combination with areas that allowed for a reliable decoding of stimuli (deepfake or natural), such as AC. An area that shows activity related to successful rejection of deepfake voices (but not natural voices) for example might indicate specific sensitivity to deepfake voices.

Reviewer #3 (Remarks to the Author):

This is a well-written, well-presented, comprehensive manuscript that provides new evidence about how the brain processes information about real vs. deepfake voices.

This is an important area for investigation given that 1) the results could inform what is distinct and socially relevant in real voices and 2) the ability to distinguish real from deepfake voices is imperative.

The investigators document that while central acoustic features are similar between real and deepfake voices, other acoustic features (vocal timbre) differed, which provides an opportunity to distinguish between real and deepfake.

Interestingly, relatively response profiles of the NAcc distinguished natural versus deepfake conditions. Meanwhile, there was no difference in NAcc activity in the speech task. This provides support for the specificity of a fingerprint of deepfake recognition within the human brain during perception.

Overall, this is a beautiful manuscript and an important contribution to the literature. It is well written and employs a thorough statistical analysis of the fMRI task condition. Though I do have several comments / questions which I feel would add to the manuscript.

Were there any laterality differences in temporal voice area regions (STP and STG/S) for voice localizer or between natural and deepfake conditions?

#Recent work by iEEG groups (i.e. Rupp et al., PLOS Biology 2022) suggest that voice encoding occurs rapidly and not all of the dynamics are resolved by functional MR measures. Given your findings in STG/S (TVAs) is it possible that greater differences in physiologic activity patterns would be identified using iEEG?

#The relationship between activity in the temporal pole and NAcc is interesting. Prior work (see Abel TJ, Rhone A, et al., 2015 J Neurosci) identify overlapping activation regions in TP cortex for faces and voices. When you examined the temporal pole region with fMRI - was there any difference between deepfake and real voices? Also, given signal dropout from the temporopolar regions - would you have been able to detect these differences?

#Target utterance followed by twelve test utterances, each of which was followed by a participant response. In the natural voice condition, target and test sentences were uttered by natural voices. In the deepfake condition, participants matched deepfake test utterances to a natural target utterance. Please justify your reasoning for not including a condition in which performance for deepfake test utterances were matched to deepfake target utterances. Given this mismatch in the relative presence of natural versus deepfake matching pairs, how does your analysis approach control for this?

Given participants underwent audiovisual training of natural identities only, how have you determined that the apparently specific activity of the NAcc is not a result of familiar identity versus unfamiliar identity? Please provide a framework for how NAcc activity as a node in the

reward network is rationalized to explain identity matching above a neural representation of familiarity given the training set matched the target set for both tasks.

#Furthermore, the experiment design section says that participants were introduced to the concept of deepfakes prior to completing the fMRI task. Have you considered the possibility that the finding of NAcc as a predominant response pattern driving differences between conditions is due to an internal encoding of reward for noticing the subtle differences between natural and deepfake voices? Given the described statistical differences between stimulus acoustic feature sets I have to assume that they sounded noticeably different. Perceptual ratings of the stimulus sets as real or not-real may be a helpful addition to aid the validation of the synthetic stimuli.

A task debriefing session is described in the methodology. Were there any relevant insights from participants provided in this session other than how they felt their focus was? I think it is relevant given the disclosure that deepfakes were discussed prior to the experiment to know if participants were primed to expect that some sounds were artificially generated.

#Please clarify whether participants were presented with the familiar identity face and names for identity fMRI task, speech matching task, or for both. Further, were faces presented for the entire test set presentation, following target presentation?

Regarding fMRI acquisition and analysis ...

Prior research has robustly demonstrated that motion $>0.5\text{mm}$ is associated with global artifactual contamination in rs-fMRI. How did authors quantify motion other than SPM realignment output? Was a motion threshold applied and were frames censored that crossed a given threshold? Please provide a thorough justification for the approach and the description of the parameterized motion and realignment vectors for all patients. See Circ et al 2017, Powers et al 2014

It is not clear in the methods which neuroanatomical atlas(es) were used to identify “low” and “high level” auditory cortex ROIs. Please add the relevant citation. Please also provide justification for your rationale for thresholding “maps” differently for NAcc, AC, TP (20%) than for IFG (30%) to define volumetric ROI analysis.

Finally please provide sequence parameters for fMRI acquisition, as well as details regarding the equipment used for in-scanner audio presentation.

Overall, this is an interesting and important paper

We would like to thank the three reviewers for their critical and helpful comments. We have addressed them point by point below (reviewers' comments in *italic*, authors' response in blue font, changes to text in blue **bold font**). We additionally uploaded a revised manuscript versions including all changes asked by the reviewers.

Reviewer #1

Comment 1

Given the unnaturalness of the deepfake voices the results are as one would expect. Performance drops for the unnatural voices. The brain, to an extent, picks up on that "unnaturalness" (or strangeness). The identity and speech task are not matched for difficulty (speech task easier) and so performance on the identity task is affected but the performance on the speech task is not.

Authors' response:

We thank the reviewer for pointing us to a more careful discussion on the relationship between task difficulty and brain response. Statistically, performance between the natural identity and the natural speech task is significantly different (Percent correct [only natural voices] \sim Task + (1|VP): estimate: 3.09, CI[1.34;4.84], p value: .001). However, we would like to highlight that the performance in both the identity and the speech task was high (identity: 86.98% (+-6.64), speech: 90.07 (+-5.02)) and the difference was only 3.09%. We argue that a performance difference of 3.09% has little relevance for behavioural research, even it is getting significant. Also, six participants even perform better in the natural identity task than in the natural speech task (see Supplementary Fig. 2). Thus, we argue that, from a perceptual point of view, the baseline identity and speech tasks are matched for task difficulty. Therefore, we think it is unlikely that the behavioural and neuronal difference observed for the deepfake identity task are predominantly modulated by task difficulty but more by the sound manipulation introduced by the deepfake generation (with more influence on identity than speech processing). Also, our brain results argue against a linear brain-behaviour relation. We observed different patterns of behavioural performance

and brain response in the NAcc for the multivariate decoding analysis (i.e. correct classification of only natural identity sounds versus high performance in the natural identity and natural and deepfake speech task), and for the univariate analysis in the AC (i.e. significant task x sound condition for task performance versus non-significant task x sound condition for univariate brain responses).

We now added a new section on the behavioural difference in the natural identity and natural speech task to the results section. The discussion on the brain-behaviour relationship was already included in the discussion:

p.6ff: **“The next step was to test whether performance differed in the identity and speech tasks with natural voices. The natural voice tasks served as baseline conditions to determine whether the difficulty of the both tasks was comparable. Statistically, performance in the natural speech task was superior to performance in the natural identity task (Percent correct [only natural voices] ~ Task + (1|VP): estimate: 3.09, CI[1.34;4.84], p value: .001). However, relative performance was high in both tasks (natural identity: 86.98% (+-6.64), natural speech: 90.07 (+-5.02)) and the difference in percentage correct was only 3.09%. Moreover, six participants performed better in the natural identity compared to the natural speech task (see Supplemental Fig. 2). Given the marginal difference in percent correct and the participants who performed better in the identity task, we argue from a perceptual perspective that the difficulties of the natural identity and speech task were relatively well matched.”**

Comment 2

In the identity matching task, natural voices elicited activity in the nucleus accumbens which the authors interpret to mean that the brain prefers natural voices because they may be more socially rewarding. Note that this pattern of activation was not found for the speech matching task (p7) which in my opinion overthrows their interpretation. The role of the basal ganglia in auditory and, more specifically, language perception and learning, is relatively well studied (e.g. Krishnan et al., 2016 TICS). The author’s interpretation re: the role of the NAcc in their experiment is post-hoc. There is nothing in the data to confirm that what is going on is indeed related to “reward processing” regarding the natural vs “fake” voices in one task but not another.

Authors' response:

We thank the reviewer for recommending a more comprehensive report and discussion on the NAcc findings. The univariate brain response in the NAcc exhibited an increase, in comparison to the deepfake identity task, during the natural identity task and across both natural and deepfake speech conditions, all of which demonstrated high matching performance. These univariate results suggest that the NAcc is responsive to stimuli deemed rewarding in the context of the ongoing task. Our interpretation of the NAcc response pattern, particularly in relation to behaviour gating to motivational or rewarding stimuli, is grounded in a substantial body of well-established findings (e.g. Knutson, Greer, 2008, Oldham et al., 2018, Mori, Zatorre, 2024 preprint). Importantly, the sensitivity observed in the NAcc does not appear to be a generic response to general reward associated with high task performance; rather, it specifically reflects sensitivity to natural identity information. The multivariate findings reveal that the NAcc accurately classifies only natural identity sounds and not speech sounds, even when the latter are also associated with high task performance. We now rewrote the corresponding result section and added to the discussion:

p.8: **“This specifically also meant that the data showed increased NAcc activity for the identity task presenting natural voices, as well as the speech task presenting natural and deepfake voices. All task conditions with high matching performance. Thus, univariate findings suggest that the NAcc is sensitive to items that are rewarding for the ongoing task 23-25. During the natural identity task, voices closely resemble the identity of the previously familiarized speakers and thus seem to more successfully guide identity-matching decisions than the deepfake version of the speakers’ identity. For the speech task, natural and deepfake voices seem to convey the task relevant information equally well (see Figure 1D).”**

p.14: **“The similar pattern of task performance and the univariate NAcc response during the identity and speech task (see Fig. 1D&Fig. 2B) may suggest that the NAcc response simply reflects the level of task performance and the related reward/non-reward towards natural and deepfake voices irrespective of the task. However, multivariate findings suggest that the NAcc seems to be specifically sensitive to the identity information conveyed by natural voices.**

The NAcc correctly classifies only natural identity sounds and not natural and deepfake speech sounds, which are also associated with high task performance.”

Comment 3

There are no hypotheses guiding the work and the work is not introduced or discussed in light of a fairly large literature on voice identity perception, including voice morphing (which is not unlike the authors' current manipulation).

Authors' response:

We thank the reviewer for the comment. It's noteworthy that, to the best of our knowledge, the human processing of audio deepfakes has primarily been explored through perceptual methods, with a notable absence of studies incorporating neuroimaging. Consequently, our study is largely exploratory in nature, as we could not derive any concrete a-priori hypotheses from previous findings. We acknowledge the reviewer's observation regarding studies employing morphing techniques for voice manipulation. It is important to clarify that these studies, while relevant, pursued different research questions than ours. They used voice morphing to disentangle natural vocal identity processing from other natural voice sound processing such as basic acoustic (Andics et al., 2013, NeuroImage, Latinus et al., 2013, CurrBiol, Lavan et al., 2019, NatComm, Bestelmeyer et al., 2022, NeuroImage) and speech related processing (Kreitewolf et al., 2014, NeuroImage, Rutten et al., 2019, NatHumBehav). In contrast, our research specifically explores identity processing of natural versus synthetically cloned vocal identities. Notably, identity cloning, a key aspect of our investigation, cannot be achieved through voice morphing but requires the application of deepfake synthesis. In our previous introduction, we highlighted behavioural studies examining visual deepfakes. In response to the feedback received, we have updated the literature review, and incorporated the latest research on human audio deepfake perception into our introduction:

p.3ff.: **“Although a large research effort is underway to develop computer algorithms for automatic deepfake generation and detection, little is known about the cognitive and neural the human ability to reliably recognize socially relevant identity information in audio 7–11 and visual 12–14 deepfakes. This is an important social and societal question, as such**

deepfakes can infiltrate and disrupt social cohesion and trust within and across social groups 15. Perceptually, human detection of audio deepfake identity as fake is unstable, across different voice synthesizing algorithms 7,8,11, and even when participants were introduced to the deepfake manipulation 8. Knowledge about the neuronal mechanisms involved in the processing of cloned speaker identities is missing to date.

We are here using psychoacoustical methods to test how well human voice identity is preserved in deepfake voices. As well as neuroimaging methods to investigate the human neurocognitive system recruited when accepting (as a potential neural indicator of deepfake deception) or rejecting (as a potential neural indicator of deepfake resilience) the deepfake as a match for a natural human identity.”

Comment 4

The findings are likely specific to that particular voice conversion manipulation (sprocket) the authors decided to use. Sprocket seems to dramatically decrease the harmonicity of the manufactured voices. Decreased harmonicity has previously been linked to decreased likability, trustworthiness etc (e.g. see Baus et al. (2019)).

Authors' response:

We thank the reviewer for this comment. We acknowledge the existing association between mean harmonicity and perceptions of likability and trustworthiness in natural voices (McAllen et al., 2014; Baus et al., 2019). We now added to the manuscript:

p.11: **“Participants rated deepfake voices significantly less natural, likable, and trustworthy than the natural voices (Supplementary Fig. 3, Supplementary Table 11), which may be related to the decreased mean harmonicity in the deepfake voices^{27,28}.”**

Comment 5

(5) I suspect that ratings of naturalness, likability and trustworthiness were correlated. Is that the case? If so, how can you differentiate between any of these variables in your regression analyses (will also affect results illustrated in Fig 4).

Authors' response:

While social ratings were correlated, there remains individual variance within each dimension (see correlation coefficients in Supplementary Table 11B). The distinct information encoded in each social dimension becomes apparent when examining the regression results (Fig. 4, Supplementary Table 13). Firstly, trustworthiness ratings revealed no significant result, despite their correlation with pleasantness and naturalness ratings. Secondly, the right AC response explained naturalness but not likability ratings (Fig.4). Thirdly, naturalness and likability ratings were related to neural response in the left AC, but to different extends. Additionally, we note that we do not face the issue of multicollinearity as we run separate models for each rating dimension. Overall, given the unique results obtained for each dimension, we propose that each social dimension encodes sufficiently distinct information, even when dimensions are correlated.

We now added the corelation matrix of the social perception ratings to the supplemental material (Supplementary Table 11B) and write in the result section:

p.12: **“The different brain-behaviour regressions identified for each social dimension propose that each dimension, despite being correlated with each other (Supplementary Table 11B), encodes sufficiently distinct information.”**

Comment 6

(6) The paper misses some basic details re: scan acquisition – what scanner was used? What tesla? How many channels has the headcoil? The voxels where resampled to 2mm³ but what was the original acquisition (in the attached “reporting summary” the reported voxel size is unusually large – 2.75x2.75x3.5mm with .6mm gap but the pdf is difficult to read due to overlapping text so I may have misread)? Do the authors have the spatial resolution to draw strong conclusions about the neuroanatomical substrates of their reported activations?

Authors’ response:

We thank the review for spotting this. We now added the relevant information to the method section. Regarding the voxel size. A voxel resolution between 2 and 3 mm³ is frequently used in auditory neuroscience and has been shown to successfully image

structures of the auditory cortex as well as subcortical regions (e.g. Erb et al., 2020; Bodin et al., 2021; Preisig et al., 2022; Staib and Frühholz, 2022; Steiner et al., 2022).

p.23ff: **“fMRI data acquisition**

Structural and functional images were recorded on a 3T-Philips Ingenia with a standard 32-channel head coil. High-resolution structural images were acquired by using T1-weighted scans (field of view, 250 × 250 × 180.6 mm; matrix, 256 × 251; 301, 1.20mm overlapping sagittal slices). Functional images were recorded by using a T2*-weighted echo-planar pulse (EPI) sequence (TR 1.6s, TE 30ms, flip angle 82°; in-plane resolution 220 × 114.2mm, voxel size 2.75 × 2.75 × 3.5mm; gap 0.6mm, ascending acquisition and in anterior commissure (AC) – posterior commissure (PC) orientation) covering the whole neocortex. In the fMRI speaker speech task, 242 volumes were acquired in each run for each participant (total 968 volumes). In the functional voice localizer experiments, 407 volumes were acquired for each participant in one run. In both fMRI experiments, volumes were acquired continuously with a TR of 1.6s.”

Reviewer #2

Comment 1

I think the study would benefit from more clearly stating the conceptualization of the basis for (neural) deepfake detection. And whether such processes are thought to be specific to deepfake detection. If it was based on social relevance, I would assume this mechanism is not only involved in deepfake voice processing but is a gradual process also involved in the evaluation of natural voice familiarity. Furthermore, such a detector would fail when deepfake voices gain social relevance. I am wondering whether this interpretation is valid in this study, as participants were only familiarized with natural but not with deepfake voices which might have impacted such processes (if I understood correctly). Could a “neural deepfake detector” instead be based on the physiological plausibility of voices?

Authors' response:

We thank the reviewer for this important comment. We hypothesise that deepfake detection relies on information from different levels including low-level bottom-up sensory and more abstract top-down information. We added this conceptualization to the introduction:

p.4: **“With this control task, we were able to statistically control for potential acoustical confounds originating from the voice deepfake synthesis. This enabled us to separate bottom-up acoustic effects from more abstract top-down effects, both of which could play a role in detecting deepfake identities.”**

We agree with the reviewer that our findings are not sufficient to state that the neural network identified in the current study is *selective* to audio deepfake processing, in line with the fact that this is a very recent development in phylogenetic terms. What we show is neurocognitive *sensitivity* to audio deepfake identities. We now added to the discussion:

p.17: **“We want to clarify that our findings suggest neural sensitivity rather than selectivity to audio deepfake identities. It remains to be tested, whether the observed cortical-striatal brain network is similarly modulated by other voice modulations, whether natural or artificially induced by other techniques.”**

We further added to the discussion:

p.17: **“Also, as audio deepfakes continue to improve their sound quality, it will be intriguing to observe whether their social acceptance also increases, and whether the neurocognitive system adjusts accordingly.”**

With regard to voice familiarity, we think that the general familiarity with the voice, e.g. the gradual transition from unfamiliar to familiar, is not a relevant factor explaining our data. We argue that voices are socially relevant irrespective of the familiarity, whereby relevant for different aspects of social communication (see Lavan & McGettigan, 2023).

Although we used the term familiarity in our discussion about the NAcc findings (p.14), we were referring to a different concept. We used familiarity to refer to the match/mismatch between the neurocognitive identity representation, that has been consolidated during the familiarization task presenting natural voices, and the incoming natural/deepfake voice during the identity task. During the natural identity task, the incoming natural voices

perfectly represent the internal identity representations and thus elicit an identity match. While during the deepfake identity task the deepfake voices do not perfectly match the internal identity representations. We now rephrased the sentence in the discussion:

p.13: **“More task-specifically, we speculate that the natural test voices during the natural identity task were expected by the listener (i.e. the internal identity representation is based on natural voices and more generally humans expect voices to be natural by default) and thus rewarding for the ongoing task as the incoming natural voice perfectly matches the internal identity representation that consequently elicited NAcc activity.”**

We apologize for any confusion, but we find it challenging to connect the term "physiological plausibility," as suggested by the reviewer, to our study. Physiology encompasses a wide range of normal low and high level biological functions in living organisms, making it difficult for us to pinpoint its relevance to our research. However, we hope that our response provided above sufficiently addresses the concern raised by the reviewer.

Comment 2

The speech content task suggests that both stimuli were well perceivable, however, beyond that differences in voice processing relevant acoustics might not affect speech content processing to the same degree (e.g., they might be in the focus of attention for voice processing). Is it possible to isolate deepfake voices that were acoustically more strongly overlapping with the natural ones and compare them to those that were not? (I assume that not enough voices were tested for this comparison). Alternatively, the appropriateness and limitations of the control task might be discussed more clearly.

Authors' response:

We thank the reviewer for this important point. We acknowledge the reviewer's point regarding our selection and evaluation of acoustic properties, which are pivotal for voice identity processing, the primary focus of our study. However, it's worth noting that indexical and linguistic information in speech show overlap to a considerable degree. For instance, listeners are better able to understand words or sentences when they are produced

by familiar speakers as compared to unfamiliar speakers (e.g. Levi et al., 2011, Souza et al., 2013, Johnsrude et al., 2013, Kreitewolf et al., 2017). Furthermore, fundamental frequency and formant frequencies have demonstrated overlapping significance in indexical and linguistic speech processing (Krumbiegel et al., 2022). Therefore, we argue that, at least, some of the acoustic attributes examined in our study may have significance for both identity-recognition and speech-related tasks. Additionally, it is plausible that the voice synthesis also modulates other acoustic characteristics, which we have not analysed, and that could have an effect on the identity and/or speech task. In summary, while the speech task and our acoustic analysis are useful to control for general acoustic effects on the identity task, they may not fully elucidate the distinct effects of specific acoustic features on each task.

We appreciate the analysis concept proposed by the reviewer, which involves classifying deepfake voices into acoustically overlapping and non-overlapping categories to assess their impact on identity and speech matching. However, there are challenges to executing this idea. Apart from the limitation posed by a small number of speakers, the acoustic characteristics in deepfake voices vary across different features. Within a single speaker's deepfake version, one acoustic attribute is accurately represented while another is poorly synthesized (Fig. 1B). Consequently, implementing a classification system to distinguish between overlapping and non-overlapping deepfake voices presents considerable difficulty. Thus, we refrained from this additional analysis and added the appropriateness and limitations of the control task to the discussion:

p.12: “The acoustic comparison of both speech sets **along a set of acoustic features mainly encoding indexical but also linguistic information** 34 revealed commonalities along the mean and variation in fundamental frequency and variation in harmonicity.”

p.13: “**The interaction observed between task and voice condition suggests that the decreased performance in identity matching of deepfake voice synthesis is not solely attributed to acoustic alterations introduced during synthesis. Rather, it also involves, to some extent, task-specific top-down effects. While the speech task and our acoustic analysis are useful to**

control for general acoustic effects on the identity task, they might not fully elucidate the distinct effects of specific acoustic features on each task.”

Comment 3

About the familiarization task: this sounds like ppl were familiarized only with the natural voices. Is it possible that the neural differences reflect familiarity to a certain extent?

Authors' response:

We thank the reviewer for this comment. Indeed, participants were only familiarized with the natural speakers' voices. Given our research aim of assessing the preservation of vocal identity information in deepfake versions of natural speakers, it would have been counterintuitive to familiarize participants with the deepfake voices as well. Regarding the potential familiarization effect in our neural data, we would like to direct the reviewer's attention to our response to the first comment above. We interpret the NAcc findings during the natural and deepfake identity task within the framework of match/mismatch between the internal identity representation and the incoming voice identity.

Given that deepfake voices are designed to clone the identity of a natural speaker, we would anticipate that a deepfake version of a natural speaker's identity could trigger a match between the internal identity representation and the incoming deepfake voice, regardless of whether the deepfake voice has been previously familiarized or not. However, it appears that our deepfake voices do not perfectly capture the intended natural identity information, resulting in a mismatch between the internal identity representation and the incoming deepfake version of the natural identity.

Comment 4

P. 7 “For the functional brain data, we [...].” In the previous paragraph several regions of interest are mentioned, here it sounds like a whole brain analysis was performed. Could you clarify?

Authors' response:

Thank you very much for spotting this inconsistency. We applied ROI analysis for the connectivity and MVPA analysis, for the univariate analysis, we did whole-brain analysis. We now deleted the misleading paragraph from the result section on p.7ff.

Comment 5

P. 8 “In comparison, during deep-fake identity matching [...].” This sounds to me like it describes the networks recruited during the deepfake voice processing. However, in case I understood it correctly, the network analysis was a comparison of natural and deepfake voice processing and not a contrast with baseline (as for natural voices), I would revise this to make clear it’s not the network “per se” but the difference.

Authors’ response:

Thank you very much for this comment. We adapted our phrasing accordingly.

p.9: “In comparison **to natural identity processing**, during deepfake identity-matching processes, such neural connectivity for speaker-specific face and memory information seems to be unavailable...”

Comment 7

P. 8 “[...] speaker-specific face and memory information seems to be unavailable given the atypical deepfake identities, [...].” If I understood correctly and only the natural voices were paired with faces and names in the familiarization period, isn’t that a problem for this interpretation? This might not be a general characteristic of deepfake processing but rather of the learning in this paradigm?

Authors’ response:

This is correct, we familiarized participants with the natural speaker identities by pairing their voices with their corresponding faces and names. Through this speaker familiarization task, participants were expected to internalize cognitive representations of the natural speakers’ identities, which is supported by the high mean familiarization rate of 96.83%. Deepfake voices aim to replicate the identity information of the original natural speaker. Thus, the associated biographical information should be available for each speaker,

regardless of the voices' naturalness. In other words, the learned biographical information should generalize to the cloned identity. And therefore, we argue that our findings are unlikely to be attributed to the learning paradigm but rather to mechanisms linked with deepfake voice processing. We now add to the result section:

p. 5: “As audio deepfakes aim to replicate the identity information of the original natural speaker, the associated biographical information should be available for natural and deepfake voice identities.”

Comment 8

*P. 10 “[...] suggesting modality-invariant encoding of speaker identity in the frontal cortex [...].”
Could you clarify why the cross-condition decoding suggests modality independent processing in IFG? Is “modality” referring to deepfake vs. natural?*

Authors' response:

We appreciate the reviewer for highlighting this aspect of our phrasing, and we acknowledge its imprecision. Indeed, by "modality" we are referring to the comparison between deepfake and natural voices. We are now writing "naturalness-invariant" encoding.

Comment 9

Was there any brain activity that was related to the task performance (whether participants were deceived or not by the deepfake?). This might be interesting in combination with areas that allowed for a reliable decoding of stimuli (deepfake or natural), such as AC. An area that shows activity related to successful rejection of deepfake voices (but not natural voices) for example might indicate specific sensitivity to deepfake voices.

Authors' response:

This suggestion is indeed intriguing. However, due to our block design, we are unable to carry out a trial-wise brain-behaviour analysis. An event-related design in contrast would have enabled us to correlate trial-specific behaviours (such as correct rejections or matching of deepfake identities) with the corresponding brain responses.

Reviewer #3

Comment 1

Were there any laterality differences in temporal voice area regions (STP and STG/S) for voice localizer or between natural and deepfake conditions?

Authors' response:

We don't have specific expectations regarding lateralization effects in the temporal lobe because past research consistently found responses to voices in both hemispheres (Pernet et al., 2015; Aglieri et al., 2021; Staib and Frühholz, 2022; Robert et al., 2023; Lamothe et al., 2024). Therefore, we analysed all brain data bilaterally and present results for both hemispheres accordingly (for findings of the contrast analysis please see Figures 2 & 3, Supplementary Figure 3a, Supplementary Table 5). We observed response differences in both the left and right temporal voice area for deepfake as compared to natural voice identity matching. Peaks are located in the bilateral superior temporal gyrus; in the left hemisphere extending to the STP (Supplementary Figure 3a, Supplementary Table 5). We now added to the method and result section:

p.27: Consistent with previous studies reporting voice-sensitive response in the bilateral temporal lobe 43,59,63,64 integrated into a broader brain network 22,59,64,65, we conducted the contrast analysis at the level of the whole brain.

Comment 2

Recent work by iEEG groups (i.e. Rupp et al., PLOS Biology 2022) suggest that voice encoding occurs rapidly and not all of the dynamics are resolved by functional MR measures. Given your findings in STG/S (TVAs) is it possible that greater differences in physiologic activity patterns would be identified using iEEG?

Authors' response:

We agree with the reviewer. With greater temporal resolution, achievable through intracranial recordings, it may become possible to differentiate between basic acoustic from higher-level categorical sound processing within the TVA, as demonstrated by Rupp and colleagues. It would be intriguing to witness future studies on natural and deepfake identity processing using methods offering higher temporal resolution. This could provide more nuanced insights into where and when distinct natural versus artificial vocal characteristics are encoded within the TVA. Regrettably, we can only speculate about the type of vocal information processed within the TVA region, that is more responsive during deepfake as compared to natural identity processing. Given that this region encompasses both the STG and STP, we speculate that both acoustic feature analysis and categorical sound processing play a role in deepfake sound identity matching. We now added the reference to the Rupp and colleagues' paper to the result section:

p.8: "Unlike the NAcc activity, activity in the pAC and hAC did not survive a task-by-naturalness interaction analysis (Fig. 2B), implying more task-general effects in the AC (Supplementary Fig. 2A & 2C, Supplementary Table 5 & 6), potentially encoding the artificial and mismatched nature of the sounds 25, which might trigger in-depth analysis both on a sound feature and on an object level 27,28."

Comment 3

The relationship between activity in the temporal pole and NAcc is interesting. Prior work (see Abel TJ, Rhone A, et al., 2015 J Neurosci) identify overlapping activation regions in TP cortex for faces and voices. When you examined the temporal pole region with fMRI - was there any difference between deepfake and real voices? Also, given signal dropout from the temporopolar regions - would you have been able to detect these differences?

Authors' response:

The contrast analysis between natural and deepfake identity matching showed no difference in BOLD response in the TP. Please note that all the results from our whole-brain contrast analysis are listed in Supplementary Table 5 & 6. We apologize for the previous version of the manuscript, which incorrectly stated that we had conducted the contrast analysis with a-priori ROIs. This is not accurate; we actually conducted a whole-

brain analysis (please also refer to reviewer 2's comment #7). We have now revised this section of the manuscript accordingly (see above and page 7 & 8 in the manuscript).

Regarding the connectivity analysis, we observed functional connectivity between the TP and NAcc during natural identity matching, and between the TP and right STG when contrasting natural against deepfake identity matching (refer to Figure 2B, Supplementary Table 7 & 8). Since we obtained significant results in the TP with the connectivity analysis, we believe that signal drop-out in the temporopolar region is not a significant issue in our sample.

Comment 4

Target utterance followed by twelve test utterances, each of which was followed by a participant response. In the natural voice condition, target and test sentences were uttered by natural voices. In the deepfake condition, participants matched deepfake test utterances to a natural target utterance. Please justify your reasoning for not including a condition in which performance for deepfake test utterances were matched to deepfake target utterances. Given this mismatch in the relative presence of natural versus deepfake matching pairs, how does your analysis approach control for this?

Authors' response:

Our decision to conduct a matching task using only natural voices as targets was based on the following reasoning:

With this study, our aim is to explore how effectively natural identity information is retained in audio deepfakes and how the human perceptual and neurocognitive system processes these deepfake identities. To investigate this question, we assessed participants' ability to match the speaker identity encoded in deepfakes to the corresponding natural identity. Had we instructed participants to match deepfake to deepfake speaker identities, we would have been addressing a different research question: the general processing of audio deepfakes, without specifically examining their relation to the corresponding natural speaker identity.

Comment 5

Given participants underwent audiovisual training of natural identities only, how have you determined that the apparently specific activity of the NAcc is not a result of familiar identity versus unfamiliar identity? Please provide a framework for how NAcc activity as a node in the reward network is rationalized to explain identity matching above a neural representation of familiarity given the training set matched the target set for both tasks.

Authors' response:

While it is true that we familiarized participants with the voices of natural speakers, or natural identities, the fundamental assumption behind deepfakes is that they replicate the identity of the natural person. Therefore, the familiar identity information of the natural speaker should also be decoded in the deepfake versions. We argue that deepfake voices carry recognizable information about the speaker and are thus not entirely unfamiliar to the participants.

We now add to the result section:

p.5: “As audio deepfakes aim to replicate the identity information of the original natural speaker, the associated biographical information should be available for natural and deepfake voice identities.”

Comment 6

Furthermore, the experiment design section says that participants were introduced to the concept of deepfakes prior to completing the fMRI task. Have you considered the possibility that the finding of NAcc as a predominant response pattern driving differences between conditions is due to an internal encoding of reward for noticing the subtle differences between natural and deepfake voices?

Authors' response:

If this were the case, we would expect to see similar response differences during the natural and deepfake speech tasks. However, what we observed instead was a task-specific modulation of naturalness in the NAcc (see results of the contrast and multivariate analysis) during the fMRI matching task.

Comment 7

Given the described statistical differences between stimulus acoustic feature sets I have to assume that they sounded noticeably different. Perceptual ratings of the stimulus sets as real or not-real may be a helpful addition to aid the validation of the synthetic stimuli.

Authors' response:

We thank the reviewer for this comment and would like to highlight our perceptual data regarding the perceived naturalness, pleasantness, and trustworthiness of both natural and deepfake sounds (see Supplementary Fig. 4, Supplementary Table 11A). Across all three perceptual scales, deepfake voices were constantly rated as little natural, pleasant, and trustworthy (average score of 2 on a five-point Likert scale, see Supplementary Table 11A), whereas natural voices have been rated as pretty natural, pleasant, and trustworthy (average score of 4 on a five-point Likert scale, see Supplementary Table 11A). These findings suggest a noticeable difference in social perception between deepfake and natural voices. However, despite these perceptually differences, identity matching between deepfake and natural voices was correct in 69% of cases, significantly surpassing the chance level of 50%. These results indicate that while audio deepfake synthesis disrupt social perception to a noticeable extent, it still preserves identity information to a degree that allows above-chance identity recognition.

Comment 8

A task debriefing session is described in the methodology. Were there any relevant insights from participants provided in this session other than how they felt their focus was? I think it is relevant given the disclosure that deepfakes were discussed prior to the experiment to know if participants were primed to expect that some sounds were artificially generated.

Authors' response:

We appreciate the reviewer's comment. Regrettably, the debriefing did not include any questions regarding differential strategies for the natural and deepfake tasks. However, we want to emphasize that throughout the entire preparation phase and the experiment itself, the concept of audio deepfakes was never introduced. Instead, during the

preparation, familiarization, and main experiment, participants were informed that they would listen to voices from four different speakers, which might sound slightly different. Given that we did not introduce the concept of deepfake voices during the experiment, we believe it is unlikely that participants were primed to expect artificial voices during the fMRI experiment. We now added to the method section:

p.21: **“In preparation for the fMRI matching task, participants passively listened to one block of sentences per speaker identity, spoken either by the natural or the deepfake voice. This step aimed to reduce any potential perceptual “surprise” effect during the main fMRI task, especially in the deepfake task conditions. A block included four different sentences, with sentence repetitions across voice-condition blocks. The order of blocks was randomized across participants. Along with the voice, we displayed the picture of the speaker’s face and the name on the screen. Importantly, we did not instruct participants that they would listen to deepfake voices of the natural speakers, instead we stated that they would hear voices that might sound slightly different from those of the natural speakers.”**

Comment 9

Please clarify whether participants were presented with the familiar identity face and names for identity fMRI task, speech matching task, or for both. Further, were faces presented for the entire test set presentation, following target presentation?

Authors’ response:

During the speaker familiarization task, we presented the speakers’ voices along with the faces and names of the speakers. However, during the fMRI task, no visual stimuli were presented with the voices, participants were only listening the respective sounds. To emphasize this auditory only design during the fMRI matching task, we added to the method section:

p. 22: **“During the entire fMRI matching task, participants exclusively listened to sounds, no face pictures or names were presented.”**

Comment 10

Prior research has robustly demonstrated that motion >0.5mm is associated with global artifactual contamination in rs-fMRI. How did authors quantify motion other than SPM realignment output? Was a motion threshold applied and were frames censored that crossed a given threshold? Please provide a thorough justification for the approach and the description of the parameterized motion and realignment vectors for all patients. See Circ et al 2017, Powers et al 2014

Authors' response:

We thank the reviewer for this comment and the information on motion correction in rsfMRI and clinical populations. Typically, for functional MRI studies investigating neurotypical individuals, motion thresholds for manual motion checks are related to the sampled voxel size. In our study, we sampled voxels with a size of $2.75 \times 2.75 \times 3.5\text{mm}$. To ensure data quality, we realigned the functional data to the AC-PC axis using the SPM realignment function, which is a robust motion correction method. Following SPM realignment, we manually checked that motion parameters were below 2.75 mm for each participant. Finally, we added the six motion correction parameters as covariates of no interest into the SPM design matrix. We have now included the missing information on the manual motion check into the method section:

p.26: “For motion correction, the functional data were first manually realigned to the AC-PC axis. Following SPM realignment, we manually inspected individual motion parameters, ensuring none exceeded 2.75mm, corresponding to the sampled voxel size.”

Comment 11

It is not clear in the methods which neuroanatomical atlas(es) were used to identify “low” and “high level” auditory cortex ROIs. Please add the relevant citation.

Authors' response:

We used the cytoarchitectonic probabilistic maps and the corresponding labelling of the STG subregions published by Zachlod and colleagues in 2020. They classified the STG into the following subregions: temporo-insular, primary, auditory secondary, higher, and

STS regions (see Figure 2 in the paper by Zachlod et al., 2020). We would like to note that we have cited this reference in the previous version of the manuscript.

Comment 12

Please also provide justification for your rationale for thresholding “maps” differently for NAcc, AC, TP (20%) than for IFG (30%) to define volumetric ROI analysis.

Authors’ response:

We employed subjective thresholds to confine the probabilistic maps to anatomically meaningful regions. See figure below showing the thresholded anatomical maps used in the study:

Comment 13

Finally please provide sequence parameters for fMRI acquisition, as well as details regarding the equipment used for in-scanner audio presentation.

Authors’ response:

Completed, and we apologize for the oversight of omitting these details in the previous version of the manuscript.

p.23ff: **“fMRI data acquisition**

Structural and functional images were recorded on a 3T-Philips Ingenia with a standard 32-channel head coil. High-resolution structural images were acquired by using T1-weighted scans (field of view, $250 \times 250 \times 180.6$ mm; matrix, 256×251 ; 301, 1.20mm overlapping sagittal slices). Functional images were recorded by using a T2*-weighted echo-planar pulse (EPI) sequence (TR 1.6s, TE 30ms, flip angle 82° ; in-plane resolution 220×114.2 mm, voxel size $2.75 \times 2.75 \times 3.5$ mm; gap 0.6mm, ascending acquisition and in anterior commissure (AC) – posterior commissure (PC) orientation) covering the whole neocortex. In the fMRI speaker speech task, 242 volumes were acquired in each run for each participant (total 968 volumes). In the functional voice localizer experiments, 407 volumes were acquired for each participant in one run. In both fMRI experiments, volumes were acquired continuously with a TR of 1.6s.”

REVIEWERS' COMMENTS:

Reviewer #1 (Remarks to the Author):

Review of revision:

I want to thank the authors for satisfactorily answering all my concerns. I have just two small suggestions:

In the abstract: "as well as a preference for socially rewarding natural speaker identities (nucleus accumbens)". This sounds like this conclusion was made from the data. Could you make clearer that the interpretation is based on previous literature by saying e.g. "previously related to socially rewarding".

In the abstract "neurocognitive mechanisms for deepfake detection". I find this statement a bit too strong, as the authors agree that such network is not claimed to be deepfake selective, but may similarly be involved in natural voices depending on the conditions, calling it a "deepfake detector" seems too much.

Reviewer #2 (Remarks to the Author):

The authors offer a thorough response to my comments in both the rebuttal letter and the manuscript. This provides important improvements to the manuscript. In my view, this manuscript is now ready for publication.

We would like to thank the reviewers for their positive evaluation of our revised manuscript. Here, we have addressed the two open points by reviewer 1 (reviewers' comments in *italic*, authors' response in blue font, changes to text in blue **bold font**). We additionally uploaded a revised manuscript versions including the changes asked by the reviewer.

Reviewer #1

Comment 1

In the abstract: "as well as a preference for socially rewarding natural speaker identities (nucleus accumbens)". This sounds like this conclusion was made from the data. Could you make clearer that the interpretation is based on previous literature by saying e.g. "previously related to socially rewarding".

Authors' response:

We thank the reviewer for this comment. We now write:

p.2: **“as well as natural speaker identities (nucleus accumbens), which are valued for their social relevance.”**

Comment 2

In the abstract "neurocognitive mechanisms for deepfake detection". I find this statement a bit too strong, as the authors agree that such network is not claimed to be deepfake selective, but may similarly be involved in natural voices depending on the conditions, calling it a "deepfake detector" seems too much.

Authors' response:

We thank the reviewer for this recommendation, and we now write:

p.2: **“Humans can thus be partly tricked by deepfakes, but the neurocognitive mechanisms identified during deepfake processing open windows for strengthening human resilience to fake information.”**